# A Superstructure Mixed-Integer Nonlinear Programming Optimization for the Optimal Processing Pathway Selection of Sludge-to-Energy Technologies

**Omar Morsy [1], Farzad Hourfar [1], Qinqin Zhu [1], Ali Almansoori [2] and Ali Elkamel [1,2,*]**

1   Department of Chemical Engineering, University of Waterloo, Waterloo, ON N2L 3G1, Canada
2   Department of Chemical Engineering, Khalifa University, Abu Dhabi P.O. Box 127788, United Arab Emirates
*   Correspondence: aelkamel@uwaterloo.ca

**Abstract:** The perception of sewage sludge has increasingly changed from being a waste, which is a burden to the environment and society, to a useful resource of materials and renewable energy. There are several available technologies at different stages of maturity that aim to convert sludge to energy in the form of electricity and/or fuels. In this paper, a decision-making support tool is proposed to help in choosing the optimal pathway for the sludge-to-energy conversion from a techno-economic perspective. The conversion technologies under study are: (1) anaerobic digestion, (2) pyrolysis, (3) gasification, (4) incineration, (5) supercritical water oxidation, (6) supercritical water gasification, as well as the corresponding dewatering and drying methods for each technology. Different synergies between the available technologies are compared by the formulation of a superstructure optimization problem expressed in a mixed-integer non-linear program (MINLP) model. The applicability of the proposed model is explored via a case study for a hypothetical sludge treatment plant with a capacity of 100 tons of dry solids (tDS) per day. The model is solved via the BARON solver using GAMS software within a reasonable processing time. According to the obtained results, the fast pyrolysis technology, coupled with filter press dewatering and thermal drying as pre-treatment steps, show the most promising outcomes with the minimum treatment cost of USD 180/tDS. Fast pyrolysis converts the sludge to bio-oil, which can be used as an alternative fuel after further refining, and biochar, which can be used for soil amendment or adsorption purposes. The model parameters are subject to uncertainty that is addressed in the sensitivity analysis section of this paper. Moreover, the pyrolysis pathway shows a high degree of robustness in most of the sensitivity analysis scenarios. Meanwhile, anaerobic digestion coupled with fast pyrolysis demonstrates the best energy recovery performance upon increasing electricity prices.

**Keywords:** sludge-to-energy; mixed-integer nonlinear programming; wastewater treatment; pyrolysis; optimization

## 1. Introduction

### 1.1. Motivation

Wastewater treatment plants (WWTPs) have been a crucial element of maintaining the health and environment of modern societies. However, these facilities require a significant amount of energy and operational costs. It has been estimated that WWTPs account for 3% of the total electricity consumption in the United States [1]. The treatment and handling of sewage sludge, which is the solid byproduct of WWTPs, accounts for approximately 30% of this electricity consumption [2] and 50% of the annual operating costs of a WWTP [3]. In addition, 73% of the treated sludge is eventually either landfilled or sent for land applications [4], which impose regularly updating stringent disposal requirements. Thus, the need for cost-effective, energy-efficient, and sustainable methods of sludge handling has recently become an active research field.

In the past decade and coinciding with the efforts to combat global warming and climate change, there has been a paradigm shift taking place towards sludge. It has shifted from being perceived only as waste and burden to society and the environment, to being looked at as a useful resource of materials and renewable energy. Several studies in the literature [5–9] reviewed available and potential technologies for energy recovery from sewage sludge in the form of electricity, heat, and/or fuels. These energy products can help in offsetting the energy consumption of the wastewater treatment facilities and thus reducing their carbon footprints as well as generating a revenue stream from products that can be sold in the market. Yet, there have been few efforts put into developing frameworks that quantitatively compare those sludge-to-energy alternatives from an economic perspective.

On the other hand, for relatively similar feedstock materials such as biomass, microalgae, and municipal solid wastes, superstructure optimization approaches have been successfully used [10–16]. Therefore, the purpose of this paper is to first provide a brief overview of a set of the most promising sludge-to-energy conversion technologies. Afterward, a mathematical model is developed, using a superstructure optimization-based approach as a decision-making support tool. Clearly, the proposed solution is useful for both researchers as well as stakeholders in municipalities looking forward to implementing master plans and strategies for biosolid handling for a sustainable future.

## 1.2. Background Information

### 1.2.1. Sludge Characterization

Sewage sludge is composed of a complex series of microorganisms, organic and inorganic solid compounds (total solids) that coexist in water heterogeneously. The organic compounds, commonly called volatile solids (VS), originate from several sources such as fecal material, plants, paper, and oils. They contain a variety of complex molecular structures from polysaccharides, lipids, proteins, and peptides to plant macromolecules, and micropollutant organic compounds such as dibenzofurans and polycyclic aromatic hydrocarbons (PAHs) [17]. The energy recovery potential in the sewage sludge is highly dependent on the amount of VS present in the sludge (i.e., the higher the percentage of volatile solids, the higher the energy content of the sludge) [18]. The inorganic compounds, also referred to as ash, are mainly composed of minerals such as silica (quartz), calcites, or microclines. Trace amounts of heavy metals are also present in sewage sludge, and examples are chromium, copper, nickel, zinc, mercury, cadmium, and lead [19]. Finally, nutrients in the form of nitrogen, potassium (potash), and phosphorus are found in the sludge and are one of the main criteria upon which the suitability of the treated sludge for usage as a fertilizer or soil conditioner depends.

### 1.2.2. Anaerobic Digestion

Anaerobic digestion is the most common process to stabilize sewage sludge in today's market [20]. In this process, a portion of the biodegradable organic compounds in the sludge is decomposed in an oxygen-free environment to a methane-rich gaseous mixture called "biogas" [17]. The unconverted portion of the organic compounds in the digester together with the inorganic compounds and moisture exit the process and are named "digested sludge" or "digestate".

The digestion process takes place in a series of complex biochemical reactions. Hydrolysis converts insoluble and high molecular weight organic compounds such as polysaccharides, proteins, and lipids into soluble amino and fatty acids. Those soluble compounds from hydrolysis are additionally split to form volatile fatty acids in the acidogenesis step. Acetogenesis is the step in which the organic acids and alcohols generated in acidogenesis are converted to acetic acid together with hydrogen and carbon dioxide. Finally, the methanogenesis step is where methane gas is predominantly produced by two different methanogenic groups of bacteria: one of them decomposes acetate to $CH_4$ and $CO_2$ and the other group utilizes $H_2$ as an electron donor and $CO_2$ as an acceptor to produce $CH_4$ [21]. The hydrolysis step is generally deemed as the rate-limiting one.

### 1.2.3. Incineration

Incineration is a process in which waste combustion takes place in a controlled manner producing flue gas, ash, and heat that can be recovered. Incineration and combustion of sewage sludge are sometimes used interchangeably; however, it needs to be noted that there is a subtle difference between both terms. Combustion is a more general term that refers to a thermochemical exothermic reaction between excess oxygen and organic material of a fuel that is completely oxidized to $CO_2$ and $H_2O$ at high temperatures. Incineration on the other hand is a special case of combustion where the combustible material originates from a waste that needs to be disposed of. The main purpose of combustion is the energy recovery from the fuel in the form of heat that can then be used in steam generation which in turn can produce electricity upon passing through steam turbines, while incineration's main purpose is the destruction of the harmful material in the waste and reducing its volume upon disposal [22]. For the purpose of our work, where energy recovery is the main interest, incineration and combustion of sewage sludge will refer to the same concept.

### 1.2.4. Gasification

Gasification is another thermochemical conversion process in which the organic components of sludge are transformed in a net reducing environment to a combustible gas called syngas, while the remaining sludge constituents are converted to ash [23]. There are lots of similarities between gasification and combustion, but they mainly differ in the lower requirement of sludge moisture content fed to the gasifier (below 15 wt%) and that oxidants are present in amounts below the stoichiometric quantities required for complete combustion or oxidation [24]. Syngas or synthesis gas is a mixture that consists mainly of hydrogen (8.89–11.17 vol%), carbon monoxide (6.28–10.77 vol%), lower percentages of methane (1.26–2.09 vol%), and C2s (0.75–1.2 vol%), along with $CO_2$ and the gasification medium [25]. The gasification medium, also called the gasifying agent, is the fluid which reacts with the sludge carbonaceous components to partially oxidize them to syngas. Typically, air with oxygen amounts of 20–40% less than that required for complete combustion is used as a gasifying agent. Nevertheless, the following media have been also studied and used in sludge gasification: pure oxygen, steam, steam–air mixture, steam–$O_2$, steam–$CO_2$, and pure $CO_2$ as reported in [22]. The gasification medium has a significant impact on the composition and accordingly the heating value of the produced syngas ranges from 4 to 12 MJ/Nm$^3$, where the highest values are obtained from gasification with pure oxygen [24]. Steam gasification increases the yield of $H_2$ in the syngas mixture compared to CO which can be attributed to both the reforming of methane and the water–gas shift reaction promoted by steam. Higher $H_2$/CO ratios correspond to higher syngas calorific values as well [26].

### 1.2.5. Pyrolysis

Pyrolysis is a thermochemical process in which the organic components of the sludge are destructed at temperatures between 300 °C and 700 °C in an oxygen-free environment [22]. Unlike combustion, which is an exothermic process, pyrolysis requires a significant amount of heat (in the range of 100 MJ/tDS) for its reactions to occur [27]. It also has a much lower moisture content tolerance to the sludge that enters the reactor (<10 wt%) and thus requires a higher drying energy [24]. The first step of the process takes place when the sludge is heated to temperatures in the range of 100–200 °C, where the remaining moisture associated with the sludge is evaporated and volatile gaseous products start to form, leaving a solid residue with non-volatiles referred to as char. These products are the result of several bond-breaking and forming reactions and are called primary pyrolysis. This is the same initial step in other thermochemical processes discussed as combustion and gasification [28]. With further heating, the next step, called secondary pyrolysis, takes place at temperatures close to 600 °C where the volatile gaseous products undergo further decomposition into simpler low molecular weight gases and stable aromatic compounds.

The vapor product is then sent for cooling and is separated into a liquid product called bio-oil and non-condensable gases (syngas).

### 1.2.6. Supercritical Water Treatment Methods

The thermochemical sludge treatment methods discussed so far, i.e., incineration, gasification, and pyrolysis, all require a drying step before the main sludge processing. The fact that raw and/or digested sludges have a significantly high moisture content, makes those processing routes rather more capital and energy intensive. An innovative way to stabilize sludge while eliminating the need for a pre-drying step is to treat it in the supercritical water (SCW) phase [29]. Supercritical water is a phase that takes place when critical temperature and pressure values of water exceed 374 °C and 22.1 Bar, respectively [29]. At such a state, one cannot distinguish between water in its liquid and vapor phase (steam) and water has unique properties. In this section, two SCW treatment methods are briefly discussed, namely supercritical water oxidation (SCWO) and supercritical water gasification (SCWG).

### Supercritical Water Oxidation (SCWO)

SCWO occurs at high temperatures and pressures (around 600 °C and 25 Bar), conditions that are well suited for the disintegration of sewage sludge [17]. Much higher oxidation rates are observed in supercritical conditions compared to subcritical ones, which can aid in the complete destruction of organic constituents of the sludge [30]. Organic compounds are mainly composed of carbon, hydrogen, nitrogen, sulfur, and phosphorus, which are oxidized to $CO_2$. $H_2O$, $N_2$, $SO_4^{2-}$, and $PO_4^{3-}$, respectively, while heavy metals are oxidized to their respective oxides [17]. Most of the oxidation reactions occur at a conversion rate of 99.9% and reaction times of 30 s or less at a temperature of 600°C, which results in relatively small reactor dimensions [31]. Another advantage of SCWO compared to incineration is the simple treatment required for the off-gas released, which is a major cost in incineration plants. Since SCWO is an exothermic reaction, energy recovery can be achieved either from heat exchange with the reactor vessel directly, or with its effluent product to produce steam [17].

### Supercritical Water Gasification (SCWG)

Similar to conventional gasification, SCWG decomposes the organic constituents of the sewage sludge into a gaseous mixture called syngas. However, the composition of the syngas from SCWG is much richer in hydrogen, which makes this technology especially attractive. Supercritical water gasification (SCWG) of sewage sludge has been studied in several research works for the purpose of hydrogen production. This technology has not been implemented yet at full scale but shows great potential for future adoption. Some of the main advantages of SCWG of biomass in general, which apply to sewage sludge as well, are as follows [32]:

- No need for prior drying of the feedstock to the SCWG reactor. Conversely, the moisture content of the feed is necessary for the reaction;
- Higher yield of $H_2$ compared to CO in the syngas product whereas in dry gasification processes CO is the main constituent of syngas and an extra water–gas shift process is required to achieve such high $H_2$:CO ratios;
- Lower amounts of coke and tar formation;
- Salts remain in the aqueous solution which avoids corrosion problems during the treatment of the produced gas.

Depending on the production scale, the hydrogen product from SCWG can be sold in the market as fuel for $H_2$ fuel cells, used in refineries, or other industrial uses (ammonia, methanol, etc.) [33].

### 1.2.7. Dewatering and Drying

The water content removal is an essential step in any sludge treatment plant to achieve a volume reduction in the stabilized product for further disposal or treatment. Such a reduction has a significant effect on the transportation and/or energy costs. There are four different categories of water/moisture present in sewage sludge: free water, adsorbed water, capillary water, and cellular water. Free water is the easiest of them to remove and is achieved by simple flotation or gravitation methods. Adsorbed and capillary waters on the other hand require much higher forces compared to free water. These higher forces can be accomplished mechanically by dewatering equipment such as centrifuges or filter presses, or chemically by the employment of flocculants. A final product called "cake" or "dewatered sludge" with a concentration greater than 30% dry solids (DS) can be achieved. This product has a semi-solid appearance and compatibility with a belt conveyor transfer or manipulation of spades. The removal of the three categories of water discussed so far can result in a volume reduction in the range of 90–95% to an effluent originally at 2% DS. The last category, cellular water, is the hardest to remove and requires even higher forces that can only be achieved thermally. Thermal dryers can produce a granular product with up to 95% DS in an efficient manner [34].

The water removal steps which lie within the scope of this study are dewatering and thermal drying. Prior to sludge dewatering, an important pre-treatment is required referred to as sludge conditioning. This step is crucial in impacting the efficiency and ease of sludge dewatering and can be achieved via different methods: thermal pre-treatment, or the use of organic and/or inorganic chemicals. The most popular chemical conditioners are inorganic lime and ferric chloride and organic polymers. Chemical type and dosage rates depend on the sludge characteristics and dewatering method/equipment type. The most common dewatering methods are belt presses, centrifuges, vacuum filters, plate, or diaphragm filter presses, and exclusively for digested sludges, sludge lagoons and drying beds [35].

On the other hand, thermal drying can be achieved either by direct or indirect methods, where the difference lies in whether the heating medium is in direct contact with the sludge or not. Direct drying methods are more commonly used. Examples of direct dryer technologies are rotary dryers, fluidized bed dryers, and belt dryers. One of the advantages of thermal drying is that it acts as both a further stabilization and volume reduction method of the sludge. The end product can be sold as Class A biosolids (pathogen-free), which are used in agricultural applications such as fertilizers. However, the high operating costs associated with drying are usually not offset by the revenues generated from selling the dried product [36]. Moreover, another problem associated with sludge drying is the potential production of odors and volatile organic compounds (VOCs) [37].

### 1.3. Sludge-to-Energy Fundamental Concepts

### 1.3.1. Sludge-to-Energy Decision-Making Frameworks

There are significant efforts being made in relation to the development of decision-making support frameworks or tools that help in ranking different sludge-to-energy alternatives. Multi-criteria decision-making (MCDM) methodologies have been applied to the problem of sludge management in [38,39]. The former study is based on traditional grey relational analysis (GRA) modified to allow for linguistic inputs, while the latter study is based on Dempster–Shafer theory and fuzzy best-worst method. Both studies consider environmental, technological, social, and economic criteria. Tang et al. proposed another MCDM framework for prioritizing different sludge technologies using four different methodologies combined with triangular fuzzy numbers to deal with hybrid-data types [40]. This work also contains a recent review of other related studies in the area of decision-making for sustainable sewage sludge management. Although MCDM tools can be useful, they are not flexible in assessing and synthesizing innovative combinations of various technologies at different capacities to maximize economic or environmental benefits. In addition, many of these tools rely on "experts' opinion", which might lead to more subjective or biased results. A more suitable approach to address those limitations would

be to formulate an optimization of mathematical models for superstructure mapping of the different alternatives. Typically, these optimization problems are modelled and solved by mixed-integer linear programming (MILP) or mixed-integer nonlinear programming (MINLP) models.

### 1.3.2. Sludge Management Optimization Models

A few studies are available in the literature which utilized MILP in solving a sludge management-related problem. A case study in [41] compared alternatives for the thermal treatment of digested sludge in the region of Zurich in Switzerland. A multi-objective MILP was developed to find the optimal environmental performance of the following technologies: sludge mono-incineration, co-incineration with municipal solid waste (MSW), and co-processing for cement manufacturing. This study did not cover any energy recovery method other than incineration, and it also did not consider the economic performance and costs associated with the potential pathways. Moreover, in [42], a stochastic multi-objective MILP model was utilized to compare different sludge utilization pathways, namely, anaerobic digestion with thermal hydrolysis, lime stabilization, incineration, land application, and selling of Class A biosolids in the market as a fertilizer. Moreover, several utilization paths for the produced biogas were considered, such as electricity production, and upgrading to compressed natural gas (CNG). The economic performance in terms of capital and operating expenditure as well as revenue from valuable products was studied. In addition, environmental performance in terms of $CO_2$ emissions and energy costs were investigated. However, similar to the case study in [41], only a few energy recovery technologies were included in the model.

The work completed in [43] focused on the whole sludge supply chain in a certain region in north-western Europe considering the synergies between 241 WWTPs. A generic decision framework called OPTIMASS, originally created for optimizing biomass supply chains [44], was customized to fit the specific application of sewage sludge. However, only a limited number of energy recovery alternatives were included in the model with the following processing equipment/routes: thickening, dewatering, MAD, thermal drying, mono/co-incineration, and utilization in the cement industry. Another shortcoming of that study was the unavailability of the parameters used in the model due to privacy agreements. Finally, in [45] anaerobic digestion, hydrothermal liquefaction, and catalytic hydrothermal gasification pathways were compared using a multi-objective superstructure optimization methodology. The developed MILP model considered both economic and environmental aspects, while CAPEX and OPEX were assumed to have linear relations. Although more technologies were assessed in that study in comparison to the former ones, the study did not consider some of the most studied sludge-to-energy technologies such as incineration, gasification, and pyrolysis.

### 1.3.3. Waste-to-Energy Optimization Models

Aside from sewage sludge, there is some available literature on the application of superstructure optimization or mixed-integer programming methodologies to find the optimal processing pathway for energy recovery from other types of wastes. The majority of those studies are related to the different types of MSW such as plastics, metals, glass, and various other organic wastes (paper, textile, food waste, etc.). For example, a fuzzy multi-objective superstructure optimization methodology with the aim of cost-minimizing while maximizing waste reduction and electricity generation was introduced in [46]. LP and MINLP (linearized to MILP) superstructure optimization models were proposed in [14,15], respectively, with a single objective function of maximizing net profit for the selected technology pathway. The presented work in [47] was not limited to only optimal technology selection, it also considered the complete supply chain of MSW including transportation between different cities. The objective function to be optimized in the MILP formulation of that study aimed at maximizing the economic benefit while considering the incurred environmental cost because of $CO_2$ emissions. Another study in [48] looked at supply

chain optimization together with technology selection via a multi-objective MILP model. The multiple objectives were: (1) minimizing economic and environmental costs, and (2) minimizing the associated risks with the chosen pathway. The latter study also included a comprehensive list of many of the works related to MSW optimization modelling frameworks.

Poultry litter is another type of waste that decision-making tools based on optimization mathematical models have been applied to. The recent work of [16,49] studied the comparison of thermochemical valorization pathways, developing mixed-integer (non)linear fractional programming models. A parametric algorithm was proposed for linearizing the optimization models to a series of MILP problems to obtain solutions in a relatively less computationally intensive way. The first study aimed at just technology selection while maximizing the return on investment (ROI). This objective function was the source of the fractional nonlinearity of the model due to the presence of a ratio of two linear equations. The second study focused on the comparison of two pyrolysis pathways, slow and fast pyrolysis, for the valorization of poultry waste, considering multiple objectives, the first being maximizing annualized profit per unit waste and the second being minimizing the equivalent $CO_2$ emissions from the chosen pathway. This study also considered optimizing the whole supply chain including the selection of the optimal location of pyrolysis facilities in relation to the waste sources taking into account transportation costs. The proposed methodology was applied to a case study for the poultry waste supply chain in the state of Georgia in the United States.

## 2. Methodology

### 2.1. Overview

The first step in the presented methodology was to identify candidate technologies that had the ability to convert sewage sludge to energy products. This was completed according to the outcomes of a comprehensive literature review process in which the strengths and drawbacks of each technology were extracted. The second step was to develop a superstructure mapping for those various alternative technologies. Subsequently, a mathematical model formulation for the optimization problem was developed in order to aid in the selection of an optimal pathway. After that, a case study was developed to test the applicability of the model by defining all the economic and technical parameters and solving for the decision variables. Finally, a sensitivity analysis was conducted over the parameters defined in the case study to assess the impacts of inherent uncertainty on the optimal solutions. The schematic of these steps is illustrated in Figure 1. In the following subsections, each of these steps is further elaborated on in detail.

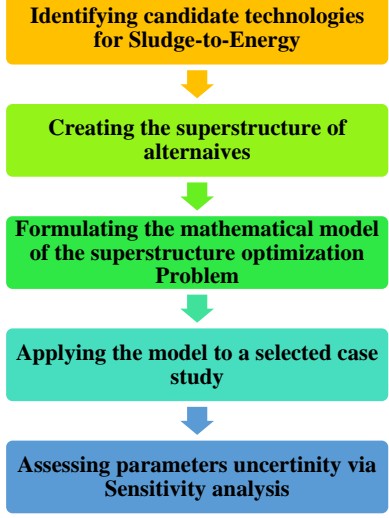

**Figure 1.** Decision-making framework for sludge-to-energy process synthesis [50].

## 2.2. Superstructure Development

The superstructure of alternatives in this work refers to a graphical representation of a network that shows the connections and relationships between the feed stream(s) being processed, potential processing technologies, and intermediate and final products. In the considered problem, there is a single feed stream crossing the boundary limit of the superstructure which is thickened sewage sludge. The processing units are categorized into biochemical processes, thermochemical processes, and intermediate processes. As shown in the schematic in Figure 2, the biochemical processes covered in this superstructure are MAD, and MAD + THP. The thermochemical processes include Incineration, Gasification, Pyrolysis, SCWO, and SCWG. The intermediate processes comprise mechanical processes such as sludge dewatering and thermal processes such as sludge thermal drying. Intermediate processes are duplicated to differentiate between those processing digested sludge and those processing undigested sludge. This is because depending on whether a biochemical technology is selected or not, intermediate processes can have varying capacities and are part of different pathways. Three different dewatering technology options were modelled, namely, belt filter dewatering, filter press dewatering, and low-speed centrifuge dewatering.

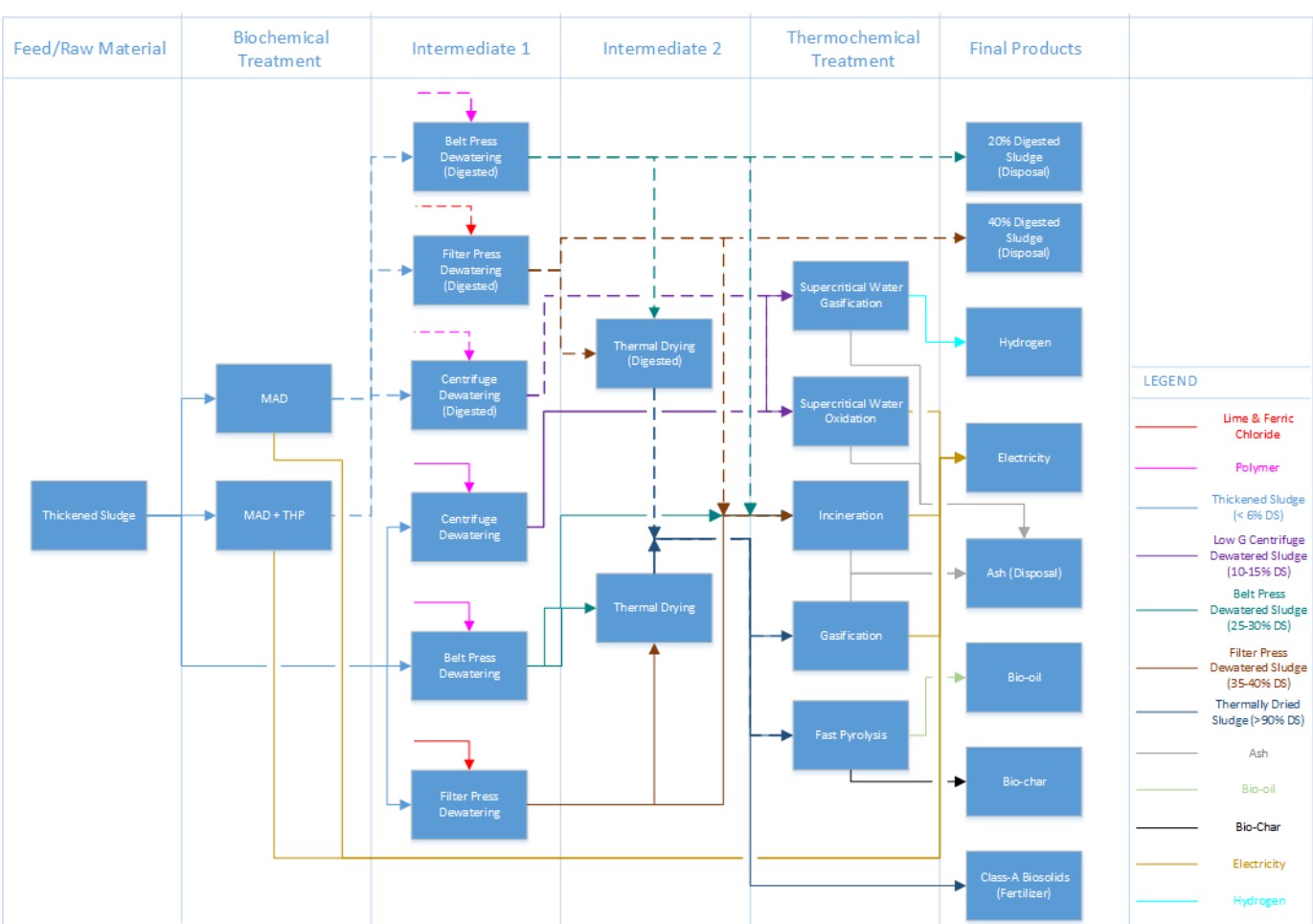

**Figure 2.** Superstructure representation of Sludge-to-Energy alternatives.

Each of the biochemical and thermochemical units includes an energy recovery facility that produces energy in the form of electricity or fuels. The final products shown in the superstructure are either value-added products or residual/waste products. Value-added products are those that can be sold in the market, such as electricity, Class A biosolids, bio-oil, biochar, and hydrogen. Residual products, such as dewatered sludge and ash, are cost-incurring ones that can be disposed into landfills or sent for beneficial use (i.e., use in cement industry for ash, land application for dewatered sludge). For ease of presentation, each material stream is given a distinct color as explained in the legend of the superstructure diagram. In addition, digested sludge products are differentiated graphically by using dashed lines compared to solid lines for undigested sludge streams. This superstructure illustration represents the foundation for the logical relationships of the building blocks of the mathematical model formulation as demonstrated in the next subsection.

*2.3. Mathematical Model Formulation*

2.3.1. General

Any optimization problem involves the minimization or maximization of a certain function, called the objective function, which is subject to a set of equality and inequality constraints. Superstructure optimization problems formulations follow the same concept and can be mathematically expressed as follows [51]:

$$
\begin{aligned}
\min_{x,z} \quad & C = c^T z + p(x) \\
\text{s.t.} \quad & r(x) = 0 \\
& s(x) + Bz \leq 0 \\
x \in R^n, \quad & z \in \{0,\ 1\}^l
\end{aligned}
\tag{1}
$$

where the objective cost function $C$ consists of: (a) costs related to a discrete decision integer variables vector $z$ which is multiplied by a matrix of relevant cost coefficients $c$ (this matrix usually consists of capital cost parameters), and (b) costs related to continuous variables vector $x$ represented in functions $p(x)$ and those are typically costs related to operation and maintenance, or revenues from product sales. The objective function is constrained by the physical performance of the process or technology efficiency, which is modelled using an equality functions vector $r(x)$, and the logical relationships are dictated by inequality functions $s(x)$ that relate to the discrete integer decision variables vector via a coefficient matrix $B$. Depending on whether functions $p(x)$, $r(x)$, and $s(x)$ are all linear or any of them are non-linear, the problem becomes a mixed-integer linear program (MILP) or mixed-integer non-linear program (MINLP), respectively, where each type has its applicable algorithms to be solved.

Equation (1) represents the generalized high-level architecture of such problems. However, in this paper, a detailed model following the same general approach, but customized to suit the specific needs of our problem, is formulated. At the beginning, the relationships between elements of the superstructure are described in this section. The proposed framework consists of a group of sets, parameters, variables, and equations. The sets are expressed by a number of bold roman letters (example: **I**), parameters use light italic roman letters (example: *I*), and variables are expressed by italic bold letters (example *X*). Model element identifiers (subscripts and superscripts) express process blocks (italic letters), streams, and their components (normal roman letters). Generic identifiers are light formatted, while if a specific identifier is used, it is **bolded**. The sets can be grouped into two main groups: sets that define the main model elements (i.e., Feed sources, technologies, process streams, components, and final products), and sets that define the relationships between those elements. The identifiers that are used to describe individual model elements that belong to a corresponding set(s) are listed in Table 1:

**Table 1.** Model element identifiers (subscripts and superscripts).

| Superstructure Element | Identifier | Description |
|---|---|---|
| | *i,j,k* | Aliases of subscripts identifiers for feed, process, and product blocks. |
| **General** | s | Generic identifier of a process stream |
| | c | Generic identifier of a component in a stream |
| **Feed Source** | *TH* | Thickened Sludge |
| | *MAD* | Mesophilic Anaerobic Digestion |
| | *MADT* | MAD + Thermal Hydrolysis Pretreatment |
| | *CD* | Centrifuge dewatering for digested sludge |
| | *CU* | Centrifuge dewatering for undigested sludge |
| | *BPD* | Belt press dewatering for digested sludge |
| | *BPU* | Belt press dewatering for undigested sludge |
| | *FPD* | Filter press dewatering for digested sludge |
| **Technologies/Processes** | *FPU* | Filter press dewatering for undigested sludge |
| | *TD* | Thermal Drying |
| | *INC* | Incineration |
| | *GN* | Gasification |
| | *PY* | Fast Pyrolysis |
| | *SCO* | Supercritical Water Oxidation |
| | *SCG* | Supercritical Water Gasification |
| | *DS20* | 20% dewatered digested sludge |
| | *DS40* | 40% dewatered digested sludge |
| | *ASH* | Ash |
| | *E* | Electricity |
| **Final Products** | *FERT* | Class A Biosolids (Fertilizer) |
| | *BO* | Bio-oil from pyrolysis |
| | *BC* | Biochar from pyrolysis |
| | *H2* | Hydrogen |
| | **THS** | Thickened Sludge |
| | **ADS** | Anaerobically Digested Sludge |
| | **E** | Electricity |
| | **P** | Polymer for chemical conditioning |
| | **L** | Lime for chemical conditioning |
| | **FC** | Ferric chloride for chemical conditioning |
| **Process Streams** | **DWS** | Dewatered Sludge |
| | **TDS** | Thermally dried sludge |
| | **ASH** | Ash |
| | **BO** | Bio-oil |
| | **BC** | Biochar |
| | **H2** | Hydrogen |
| | **VS** | Total volatile solids |
| | **ASH** | Ash |
| | **DS** | Total dry solids (VS + Ash) |
| **Components in process streams** | **W** | Water or moisture in the sludge/biosolids |
| | **E** | Electricity |
| | **BO** | Bio-oil |
| | **BC** | Biochar |
| | **H2** | Hydrogen |

Moreover, the sets describing the model elements and their relationships are described in Table 2.

**Table 2.** Sets of model elements and their relationships.

| Set | Description |
|---|---|
| **I** | Combined set of feed, process, and final product blocks |
| **FEED** | Subset of feed blocks, **FEED** $\subset$ **I** |
| **PROCESS** | Subset of processing technologies, **PROCESS** $\subset$ **I** |
| **PRODUCT** | Subset of final products, **PRODUCT** $\subset$ **I** |
| **STR** | Set of process streams |
| **CHEM** | Subset of chemicals streams used for conditioning **CHEM** $\subset$ **STR** |
| **COMP** | Set of components of process streams |
| **S**$_i$ | Set of descendant block(s) from block $i \in$ **FEED** $\cup$ **PROCESS**. Where **S**$_i$ $\subset$ **PROCESS** $\cup$ **PRODUCT** |
| **P**$_i$ | Set of precedent block(s) of block $i \in$ **PROCESS** $\cup$ **PRODUCT**. Where **P**$_i$ $\subset$ **FEED** $\cup$ **PROCESS** |
| **STF**$_i$ | Set of inlet stream(s) applicable with process $i \in$ **PROCESS**. Where **STF**$_i$ $\subset$ **STR** |
| **STPR**$_i$ | Set of outlet stream(s) applicable with process $i \in$ **PROCESS**. Where **STPR**$_i$ $\subset$ **STR** |
| **SCOMP**$_s$ | Set of component(s) applicable to stream $s \in$ **STR**. Where **SCOMP**$_s$ $\subset$ **COMP** |
| **FPCO**$_i$ | Set of component(s) used for specifying the revenue/disposal cost of a final product $i \in$ **PRODUCT**. Where **FPCO**$_i$ $\subset$ **COMP** |

After specifying the sets, defining model elements, and their relationships, a group of performance and economic parameters applicable to all the processing technologies are defined (Table 3).

**Table 3.** Parameters applicable to all processing technologies.

| Parameter | Description |
|---|---|
| $CAP_i$ | Maximum processing capacity of a certain process $i \in$ **PROCESS** in tDS/day. |
| $BCC_i$ | Base (reference) capital cost of process $i \in$ **PROCESS** in **USD** (USD 2019) |
| $BQ_i$ | Base (reference) processing capacity of process $i \in$ **PROCESS** used in capital cost calculation. |
| $\alpha_i$ | Economies of scale exponent of process $i \in$ **PROCESS**. |
| $POC_i$ | Operating cost parameter for a certain process $i \in$ **PROCESS**. |
| $DPY$ | Days of operation per year |

The next component to be defined for the model formulation is the decision variables. The variables can be grouped in several ways: process variables versus economic variables, continuous variables versus integer and/or binary variables, and dependent variables versus independent variables. In terms of the mathematical model formulation, what matters the most is the distinction between continuous and integer/binary variables, because this will play a key role in determining the type of the optimization problem and its solution. Table 4 lists the different variables that are part of the general model formulation.

**Table 4.** Variables for general model formulation.

| Variable | Type | Description |
|---|---|---|
| $FI_i^{s,c}$ | Process, continuous, dependent | Total inlet flowrate of a component c $\in$ **COMP** within a process stream s $\in$ **STR** into process $i \in$ **PROCESS**. |
| $FO_i^{s,c}$ | Process, continuous, dependent | Total outlet flowrate of a component c $\in$ **COMP** within a process stream s $\in$ **STR** out of process $i \in$ **PROCESS**. |
| $X_{i,j}^{s,c}$ | Process, continuous, dependent | Flowrate of a component c $\in$ **COMP** within a process stream s $\in$ **STR** going from any block $i \in$ **I** to another block $j \in$ **I**. |
| $SF_{i,j}^s$ | Process, continuous, independent | Split factor of a process stream s $\in$ **STR** going from any block $i \in$ **I** to another block $j \in$ **I**. |
| $z_i$ | Process, binary, independent | Binary variable that dictates whether a certain process $i \in$ **PROCESS** exists or not. $z_i \in \{0,1\}$ |
| $CC_i$ | Economic, continuous, dependent | Capital cost of a certain process $i \in$ **PROCESS** in USD (USD 2019) |
| $OC_i$ | Economic, continuous, dependent | Operating cost of a certain process $i \in$ **PROCESS** in USD/yr (USD 2019). |

The relationship between the different process variables is represented graphically in Figure 3. For a given process, the flowrate of each component c in an inlet stream s is calculated by summing all the individual flowrates of the same component and stream from the preceding blocks of that process. On the other hand, the individual flowrate of a certain component in a stream going from a certain block to a subsequent one is dictated by a split factor $SF_{i,j}^s$ ranging from 0 to 1 which is specific to each stream, origin process, and destination block. These concepts are mathematically represented in Equations (2)–(7). Equation (8) forces the total split factors originating from a certain process, which are equal to zero in case the process is not chosen. Similarly, Equation (9) forces the total sludge dry solids inlet flowrate to a certain process equal to zero in case the process is not chosen. If the process is selected, this equation ensures the flowrate does not exceed the maximum capacity. Equation (10) forces a minimum flow of 10% of the maximum capacity to enter a certain process if it is selected. The relationship between total inlet flows and outlet flows of relevant components and streams of a certain process is discussed for each block in the next sections.

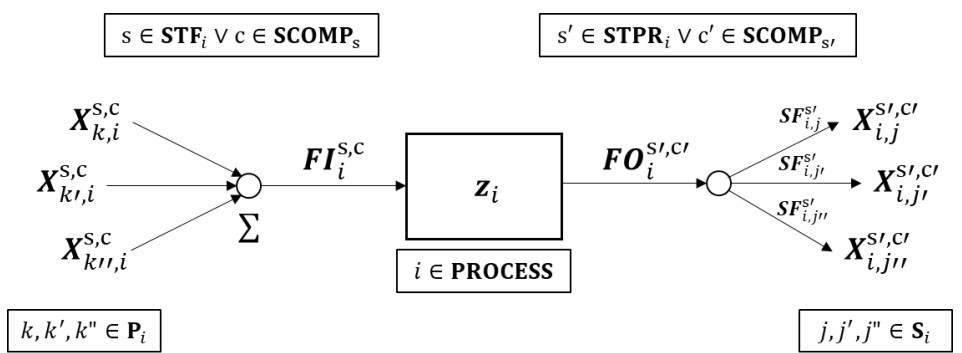

**Figure 3.** Graphical representation of relationships between the model's process variables.

$$FI_i^{s,c} = \sum_{k \in \mathbf{P}_i} X_{k,i}^{s,c}, \ \forall \ (i \in \textbf{PROCESS} \ \wedge \ \text{s} \in \textbf{STF}_i \ \wedge \text{c} \in \textbf{SCOMP}_\text{s}) \tag{2}$$

$$FI_i^{s,c} = 0, \ \forall \ (i \in \textbf{PROCESS} \ \wedge \ (\text{s} \notin \textbf{STF}_i \ \vee \text{c} \notin \textbf{SCOMP}_\text{s})) \tag{3}$$

$$FO_i^{s,c} = 0, \ \forall \ (i \in \textbf{PROCESS} \ \wedge \ (\text{s} \notin \textbf{STPR}_i \ \vee \text{c} \notin \textbf{SCOMP}_\text{s})) \tag{4}$$

$$X_{i,j}^{s,c} = SF_{i,j}^s * FO_i^{s,c}, \ \forall \ (i \in \textbf{PROCESS} \ \wedge \ j \in \mathbf{S}_i \wedge \ \text{s} \in \textbf{STPR}_i \ \wedge \text{c} \in \textbf{SCOMP}_\text{s}) \tag{5}$$

$$0 \leq \ SF_{i,j}^s \leq 1, \ \forall (i \in \textbf{PROCESS} \ \wedge \ j \in \mathbf{S}_i \wedge \ \text{s} \in \textbf{STPR}_i) \tag{6}$$

$$SF_{i,j}^{s} = 0, \ \forall \left( i \in \textbf{PROCESS} \ \wedge \left( s \notin \textbf{STF}_j, \ j \in \textbf{S}_i \right) \right) \tag{7}$$

$$\sum_{j \in \textbf{S}_i} SF_{i,j}^{s} = z_i, \ \forall \left( i \in \textbf{PROCESS} \ \wedge \ s \in \textbf{STPR}_i \right) \tag{8}$$

$$FI_i^{s,\textbf{DS}} \leq z_i * CAP_i, \ \forall \left( i \in \textbf{PROCESS} \ \wedge \ s \in \textbf{STF}_i \right) \tag{9}$$

$$FI_i^{s,\textbf{DS}} \geq 0.1 * z_i * CAP_i, \ \forall \left( i \in \textbf{PROCESS} \ \wedge \ s \in \textbf{STF}_i \right) \tag{10}$$

It should be noted that the sets, parameters, variables, and equations stated above are not conclusive of all the mathematical model formulations. So, more sets, parameters, variables, and equations specific to each block are defined in the next sections.

2.3.2. Thickened Sludge Block

The thickened sludge block represents the feed stream that is distributed among the different subsequent alternatives. Table 5 lists all the parameters that are exclusively relevant to this block.

**Table 5.** Model elements applicable to the thickened sludge block.

| Symbol | Type | Description | Units/Set Elements |
|--------|------|-------------|--------------------|
| $FTHS$ | Parameter | Flowrate of thickened sludge to be processed in tons of dry solids per day | tDS/day |
| $FPVS$ | Parameter | Feed volatile solids mass percentage of total dry solids flowrate | % |
| $FPASH$ | Parameter | Ash mass percentage of dry solids | % |
| $PDS_{THS}$ | Parameter | Dry solids mass percentage of total sludge flowrate | % |

Equations (11)–(13) define the total flowrates of the various components in the thickened sludge stream. Equations (14)–(18) define the individual flowrates of those components going to any of the applicable descendant blocks.

$$FO_{TH}^{\textbf{THS,VS}} = FPVS * FTHS \tag{11}$$

$$FO_{TH}^{\textbf{THS,ASH}} = FPASH * FTHS \tag{12}$$

$$FO_{TH}^{\textbf{THS,DS}} = FTHS \tag{13}$$

$$X_{TH,j}^{\textbf{THS, VS}} = SF_{TH,j}^{THS} * FO_{TH}^{\textbf{THS,VS}}, \ \forall \ j \in \textbf{S}_{TH} \tag{14}$$

$$X_{TH,j}^{\textbf{THS, ASH}} = SF_{TH,j}^{THS} * FO_{TH}^{\textbf{THS,ASH}}, \ \forall \ j \in \textbf{S}_{TH} \tag{15}$$

$$X_{TH,j}^{\textbf{THS, DS}} = X_{TH,j}^{\textbf{THS, VS}} + X_{TH,j}^{\textbf{THS, ASH}}, \ \forall \ j \in \textbf{S}_{TH} \tag{16}$$

$$X_{TH,j}^{\textbf{THS,W}} = X_{TH,j}^{\textbf{THS, DS}} * \frac{1 - PDS_{THS}}{PDS_{THS}}, \ \forall \ j \in \textbf{S}_{TH} \tag{17}$$

$$\sum_{j \in \textbf{S}_{TH}} SF_{TH,j}^{\textbf{THS}} = 1 \tag{18}$$

2.3.3. Anaerobic Digestion Blocks

The anaerobic digestion blocks convert the thickened sludge stream into electricity, which is exported to the grid or used onsite. Meanwhile, the digested sludge is sent to any of the available dewatering options. Table 6 lists all the model elements that are exclusively relevant to this block.

| Symbol | Type | Description | Units/Set Elements |
|---|---|---|---|
| **AD** | Set | Subset of anaerobic digestion blocks **DW** $\subset$ **PROCESS** | *{MAD, MADT}* |
| $VSD_i$, $i \in$ **AD** | Parameter | Volatile solids destruction percentage | % |
| $Y_i^E$, $i \in$ **AD** | Parameter | Yield of net electricity per ton of dry volatile solids destructed | kWh/tVSD |

Equations (19)–(21) define the yield of each component in the outlet product stream of digested sludge, while Equation (22) defines the second outlet product stream of electricity generated from biogas utilization.

$$FO_i^{\textbf{ADS,VS}} = FI_i^{\textbf{THS,VS}} * (1 - VSD_i), \quad \forall\, i \in \textbf{AD} \tag{19}$$

$$FO_i^{\textbf{ADS,ASH}} = FI_i^{\textbf{THS,ASH}}, \quad \forall\, i \in \textbf{AD} \tag{20}$$

$$FO_i^{\textbf{ADS,DS}} = FO_i^{\textbf{ADS,VS}} + FO_i^{\textbf{ADS,ASH}}, \quad \forall\, i \in \textbf{AD} \tag{21}$$

$$FO_i^{\textbf{E, E}} = FI_i^{\textbf{THS,VS}} * VSD_i * Y_i^E, \quad \forall\, i \in \textbf{AD} \tag{22}$$

Moreover, Equations (23) and (24) define the capital and operating costs of any of the anaerobic digestion blocks, respectively.

$$CC_i = BCC_i * \left( \frac{FI_i^{\textbf{THS,DS}}}{BQ_i} \right)^{\alpha_i}, \quad \forall\, i \in \textbf{AD} \tag{23}$$

$$OC_i = POC_i * FI_i^{\textbf{THS,DS}} * DPY, \quad \forall\, i \in \textbf{AD} \tag{24}$$

### 2.3.4. Dewatering Blocks

The function of dewatering blocks is to reduce the moisture content of the influent sludge after being conditioned with a certain chemical that enhances its dewaterability. Three dewatering methods are available in the superstructure, namely centrifuge, belt press, and filter press, each of which is capable of achieving a different degree of cake dryness. For each dewatering method, a distinct block is modelled depending on the type of feed sludge entering, undigested thickened sludge, or anaerobically digested sludge. The subsequent processing step/destination differs depending on the dewatering method and its feed. Table 7 lists all the sets, variables, and parameters that are relevant to this block.

| Symbol | Type | Description | Units/Set Elements |
|---|---|---|---|
| **DW** | Set | Subset of dewatering processes **DW** $\subset$ **PROCESS**. | *{CU, CD, BPU, BPD, FPU, FPD}* |
| **CH**$_i$ | Set | Set of matching a certain chemical conditioning stream $s \in$ **CHEM** to a corresponding dewatering process $i \in$ **DW**. | *{P} for i = CU, CD, BPU, and BPD* <br> *{L, FC} for i = FPU, and FPD* |
| $DR^s$ | Parameter | Dosage rate of conditioning chemical stream s $\in$ **CHEM**. | ton/tDS |
| $PDS_i$ | Parameter | Percentage of total dry solids in dewatering process $i \in$ **DW**. | % |
| $CH_i^s$ | Variable | Flowrate of conditioning chemical s $\in$ **CHEM** to a certain dewatering technology $i \in$ **DW** | ton/day |

In Equation (25), the flowrate of the relevant conditioning chemical to a certain dewatering process is defined as a function of the sludge dry solids flowrate multiplied by the dosage rate parameter. Equations (26)–(29) define the yield of each component in the outlet product stream of the dewatered sludge.

$$CH_i^s = \sum_{s' \in \textbf{STF}_i} FI_i^{s',\textbf{DS}} * DR^s, \quad \forall \, (i \in \textbf{DW} \, \wedge \, s \in \textbf{CH}_i) \tag{25}$$

$$FO_i^{\textbf{DWS,VS}} = \sum_{s' \in \textbf{STF}_i} FI_i^{s',\textbf{VS}}, \quad \forall \, i \in \textbf{DW} \tag{26}$$

$$FO_i^{\textbf{DWS,ASH}} = \sum_{s' \in \textbf{STF}_i} FI_i^{s',\textbf{ASH}} + \sum_{s \in \textbf{CH}_i} CH_i^s, \quad \forall \, i \in \textbf{DW} \tag{27}$$

$$FO_i^{\textbf{DWS,DS}} = FO_i^{\textbf{DWS,VS}} + FO_i^{\textbf{DWS,ASH}}, \quad \forall \, i \in \textbf{DW} \tag{28}$$

$$FO_i^{\textbf{DWS,W}} = FO_i^{\textbf{DWS,DS}} * \frac{1 - PDS_i}{PDS_i}, \quad \forall \, i \in \textbf{DW} \tag{29}$$

Equations (30) and (31) define the capital and operating costs of the various sludge dewatering blocks, respectively.

$$CC_i = BCC_i * \left( \frac{FI_i^{\textbf{s,DS}}}{BQ_i} \right)^{\alpha_{DW}}, \quad \forall \, (i \in \textbf{DW} \, \wedge \, s \in \textbf{STF}_i) \tag{30}$$

$$OC_i = POC_{DW} * FI_i^{\textbf{s,DS}} * DPY, \quad \forall \, (i \in \textbf{DW} \, \wedge \, s \in \textbf{STF}_i) \tag{31}$$

2.3.5. Thermal Drying Block

The thermal drying block further reduces the moisture content in the sludge using heat. A single block is modelled to receive sludge from the existing dewatering blocks. The dried sludge is routed to the possible subsequent options, namely, pyrolysis and/or being sold as Class A biosolids fertilizer. The model elements applicable to the thermal drying block are identified in Table 8.

**Table 8.** Model elements for the thermal drying block.

| Symbol | Type | Description | Units |
|---|---|---|---|
| $PDS_{TD}$ | Parameter | Percentage of total dry solids from thermal drying | % |
| $FWE_{TD}$ | Variable | Total flowrate of water evaporated in the thermal dryer | tonH$_2$O/day |

Equations (32)–(35) define the yield of each component in the outlet product stream of the thermally dried sludge. Equation (36) defines the amount of water/moisture evaporated in the dryer, which is a key parameter in sizing the dryer for cost estimating.

$$FO_{TD}^{\textbf{TDS,VS}} = FI_{TD}^{\textbf{DWS,VS}} \tag{32}$$

$$FO_{TD}^{\textbf{TDS,ASH}} = FI_{TD}^{\textbf{DWS,ASH}} \tag{33}$$

$$FO_{TD}^{\textbf{TDS,DS}} = FI_{TD}^{\textbf{DWS,DS}} \tag{34}$$

$$FO_{TD}^{\textbf{TDS,W}} = FO_{TD}^{\textbf{TDS,DS}} * \frac{1 - PDS_{TD}}{PDS_{TD}} \tag{35}$$

$$FWE_{TD} = FI_{TD}^{\textbf{DWS,W}} - FO_{TD}^{\textbf{TDS,W}} \tag{36}$$

Equations (37) and (38) define the capital and operating costs of the thermal drying block, respectively.

$$CC_{TD} = BCC_{TD} * \left( \frac{FWE_{TD}}{BQ_{TD}} \right)^{\alpha_{TD}} \tag{37}$$

$$OC_{TD} = POC_{TD} * FWE_{TD} * DPY \tag{38}$$

### 2.3.6. Incineration Block

The incineration block is modelled to have a single input which is the sludge (either digested or not), and two outputs which are net electricity generated and the residual ash. The net electricity generated is calculated in two steps. First, the heat losses from the incinerator and the heat required for moisture evaporation are both subtracted from the lower heating value of the sludge. This difference resembles the recovered heat in the waste heat boiler which generates steam. The second step is to multiply the calculated steam enthalpy by the efficiency of the Rankine cycle to obtain the produced net electricity. Model elements relevant to the incineration block are listed in Table 9. It should be noted that the LHV of the sludge can be impacted by the addition of lime as a conditioner for the filter press dewatering step. Lime stabilizes/inhibits some of the volatile solids in the sludge; the exact value of the reduction is uncertain and will be subject to sensitivity analysis.

**Table 9.** Model elements applicable to the incineration block.

| Symbol | Type | Description | Units |
|---|---|---|---|
| $LHV_{VS}$ | Parameter | Lower heating value parameter (coefficient) for sludge | MJ/tVDS |
| $\lambda_W$ | Parameter | Latent heat of vaporization of water | MJ/ton |
| $HLF$ | Parameter | Heat Loss Factor in the incinerator | Dimensionless |
| $\eta_R$ | Parameter | Efficiency of the Rankine cycle | % |
| $CF_{MJ2kWh}$ | Parameter | Conversion factor of MJ to kWh | Dimensionless |
| $H_{INC}^{VS}$ | Variable | Heat flow of volatile solids entering the incineration block | kWh(th)/day |
| $H_{INC}^{W}$ | Variable | Heat required to evaporate moisture in sludge entering the incineration block | kWh(th)/day |
| $H_{INC}$ | Variable | Net heat recovered from incineration | kWh(th)/day |

Equation (39) specifies that the yield of the ash produced out of incineration is equal to that fed from the incoming dewatered sludge. Equation (10) defines the heat content of the sludge based on its volatile solids content. Equation (41) calculates the amount of heat required to evaporate the moisture content of the sludge. Equation (43) defines the net amount of electricity which can be recovered via a Rankine cycle using the heat input calculated in Equation (42).

$$FO_{INC}^{ASH,ASH} = FI_{INC}^{DWS,ASH} \tag{39}$$

$$H_{INC}^{VS} = LHV_{VS} * FI_{INC}^{DWS,VS} * CF_{MJ2kWh} \tag{40}$$

$$H_{INC}^{W} = \lambda_W * FI_{INC}^{DWS,W} * CF_{MJ2kWh} \tag{41}$$

$$H_{INC} = \left( H_{INC}^{VS} - H_{INC}^{W} \right) * (1 - HLF) \tag{42}$$

$$FO_{INC}^{E,E} = H_{INC} * \eta_R \tag{43}$$

Equations (44) and (45) define the capital and operating costs of the incineration block, respectively.

$$CC_{INC} = BCC_{INC} * \left( \frac{FI_{INC}^{DWS,DS}}{BQ_{INC}} \right)^{\alpha_{INC}} + 1147 \left( FO_{INC}^{E,E} \right)^{0.695} \tag{44}$$

$$OC_{INC} = (POC_{INC} * FI_{INC}^{DWS,DS} + POC_{ST} * FO_{INC}^{E,E}) * DPY \tag{45}$$

where *ST* refers to a steam turbine unit for electricity generation.

### 2.3.7. Gasification Block

For modelling the gasification block, the thermal drying unit is included inside its boundaries. The reason behind this assumption is the recycling of heat from syngas combustion is utilized to both dry the sludge and provide the necessary heat for the gasifier (overall endothermic reaction). Modelling the blocks separately with recycle streams will be challenging. So, in order to simplify the problem, the units are combined together since there are available data in the literature about net electricity generated from such a configuration.

The effect of moisture in the dewatered sludge entering the gasification block on the produced net electricity is negligible at moisture contents below 80% [52]. Therefore, the model will be insensitive on whether the sludge to gasification is from belt press, or from filter press dewatering units.

The yield of net electricity obtained in [52] was for certain conditions, i.e., sludge composition, temperature, pressure, and sludge drying level. For our modelling purposes, all the conditions are assumed to remain the same for the optimized design, except for the sludge composition (%volatile solids). Accordingly, the net power produced in [52] will be divided by the amount of volatile solids entering in this study and is assumed to increase linearly with %VS. The only additional parameter to be defined that is exclusively applicable to gasification is $YF_{GN}^E$, which resembles the yield of net electricity in kWh per ton dry volatile solids fed to the gasifier.

Equations (46) and (47) define the yields of gasification products as net electricity and ash, respectively.

$$FO_{GN}^{E,E} = FI_{GN}^{DWS,VS} * YF_{GN}^E \tag{46}$$

$$FO_{GN}^{ASH,ASH} = FI_{GN}^{DWS,ASH} \tag{47}$$

Equations (48) and (49) define the capital and operating costs of the gasification block.

$$CC_{GN} = BCC_{GN} * \left( \frac{FI_{GN}^{DWS,DS}}{BQ_{GN}} \right)^{\alpha_{GN}} \tag{48}$$

$$OC_{GN} = POC_{GN} * FI_{GN}^{DWS,DS} * DPY \tag{49}$$

### 2.3.8. Pyrolysis Block

For modelling purposes, linear empirical equations were found in the literature that predict the yield of both bio-oil and biochar [20]. The yield parameters are a function of the percentage of volatile solids and total dry solids entering the pyrolysis reactor. The yield of syngas is usually not accounted for since it is of negligible heating value. Accordingly, the syngas stream is excluded from our model. The correction factors $CF_{PY}^{BO}$ and $CF_{PY}^{BC}$ are added to Equations (46) and (47), respectively, to account for the uncertainty in the coefficients of the empirical equation upon doing the sensitivity analysis for model parameters. Pyrolysis is an endothermic reaction, so the need for an auxiliary fuel exists to reach the required operating conditions. The heat duty required is calculated as the summation of the heat of drying any residual moisture, the sensible heat to reach reaction temperature, and the heat of reaction. From the calculated heat duty, the amount of required natural gas is obtained to satisfy the energy balance and also to estimate the operating cost parameters for the unit.

Equations (50) and (51) define the yields of fast pyrolysis products, bio-oil and biochar, in their respective order.

$$FO_{PY}^{\textbf{BO,BO}} = \left(63.68\% * FI_{PY}^{\textbf{TDS,VS}} - 11.34\% * FI_{PY}^{\textbf{TDS,DS}}\right) * CF_{PY}^{BO} \tag{50}$$

$$FO_{PY}^{\textbf{BC,BC}} = -78.95\% * FI_{PY}^{\textbf{TDS,VS}} + 98.79\% * FI_{PY}^{\textbf{TDS,DS}} * CF_{PY}^{BC} \tag{51}$$

Equations (52) and (53) define the capital and operating costs of the pyrolysis block, respectively.

$$CC_{PY} = BCC_{PY} * \left(\frac{FI_{PY}^{\textbf{DWS,DS}}}{BQ_{PY}}\right)^{\alpha_{PY}} \tag{52}$$

$$OC_{PY} = POC_{PY} * FI_{PY}^{\textbf{DWS,DS}} * DPY \tag{53}$$

### 2.3.9. SCWO and SCWG blocks

The SCWO and SCWG blocks are modelled in a way to simply convert the volatile solids portion of the sludge fed into the block to the energy product of each unit. Table 10 lists the yield parameters defined for each of these two processes in the model.

**Table 10.** Model parameters applicable to the SCWO and SCWG blocks.

| Symbol | Type | Description | Units |
|--------|------|-------------|-------|
| $YF_{SCO}^{E}$ | Parameter | Yield of net electricity per ton dry volatile solids fed to the SCWO block | kWh/tVS |
| $YF_{SCG}^{H2}$ | Parameter | Yield of hydrogen per ton dry volatile solids fed to the SCWG block | kgH2/tVS |

Equation (54) defines the electricity product yield from SCWO while Equation (56) defines that of hydrogen from SCWG. Equations (55) and (57) calculate the ash product yield from SCWO and SCWG, respectively.

$$FO_{SCO}^{\textbf{E,E}} = FI_{SCO}^{\textbf{DWS,VS}} * Y_{SCO}^{E} \tag{54}$$

$$FO_{SCO}^{\textbf{ASH,ASH}} = FI_{SCO}^{\textbf{DWS,ASH}} \tag{55}$$

$$FO_{SCG}^{\textbf{H2,H2}} = FI_{SCG}^{\textbf{DWS,VS}} * Y_{SCG}^{H2} \tag{56}$$

$$FO_{SCG}^{\textbf{ASH,ASH}} = FI_{SCG}^{\textbf{DWS,ASH}} \tag{57}$$

The economic variables of capital and operating costs of SCWO and SCWG are defined in Equations (58) and (59).

$$CC_{SCO} = BCC_{SCO} * \left(\frac{FI_{SCO}^{\textbf{DWS,DS}}}{BQ_{SCO}}\right)^{\alpha_{SCO}} \tag{58}$$

$$OC_{SCO} = POC_{SCO} * FI_{SCO}^{\textbf{DWS,VS}} * DPY \tag{59}$$

$$CC_{SCG} = BCC_{SCG} * \left(\frac{FI_{SCG}^{\textbf{DWS,DS}}}{BQ_{SCG}}\right)^{\alpha_{SCG}} \tag{60}$$

$$OC_{SCG} = POC_{SCG} * FI_{SCG}^{\textbf{DWS,DS}} * DPY \tag{61}$$

### 2.3.10. Objective Function

The objective function to be minimized in the optimization problem is the net annual cost defined in Equation (62), which is the difference between annual costs and annual revenues. The annual costs comprise of the total annualized capital costs (defined by Equations (63) and (64)), total annual operating costs of the optimal pathway technologies chosen by the model, as in Equation (65), and the total disposal costs of the produced

byproducts (Equation (66)). The total annual revenue from the sales of final products is specified in Equation (67). The total flow of each final product or byproduct is defined in Equation (68) and it is used in the calculation of revenue and disposal costs variables. The model elements for defining the equations related to the objective function are listed in Table 11.

**Table 11.** Model elements applicable to the objective function definition.

| Symbol | Type | Description | Units/Set Elements |
|--------|------|-------------|--------------------|
| **REVP** | Set | Subset of revenue-generating products **REVP** $\subset$ **PRODUCT** | {*E*, *Fert*, *BO*, *BC*, *H2*} |
| **DISP** | Set | Subset of cost-incurring products to be disposed **DISP** $\subset$ **PRODUCT** | {*DS20*, *DS40*, *ASH*} |
| *NETCOST* | Variable | Objective function variable to be minimized representing the net production cost of the chosen pathway | USD/yr (USD 2019). |
| *TACC* | Variable | Total annualized capital costs of the chosen processes in the optimal pathway. | USD/yr (USD 2019). |
| *TOC* | Variable | Total annualized operating costs of the chosen processes in the optimal pathway. | USD/yr (USD 2019). |
| *TADC* | Variable | Total annual disposal costs from the disposal of final byproducts. | USD/yr (USD 2019). |
| *TREV* | Variable | Total revenues from selling of final products. | USD/yr (USD 2019). |
| $FPI_i$ | Variable | Total flowrate of a final product $i \in$ **PRODUCT** | unit product/day |
| *AF* | Parameter | Annualized capital charge ratio | dimensionless |
| *d* | Parameter | Interest/discount rate | % |
| *n* | Parameter | Number of years of the project life | yr |
| $SP_i$ | Parameter | Price of selling of a final product $i \in$ **REVP**. | USD/unit product |
| $DC_i$ | Parameter | Disposal cost of a final product $i \in$ **DISP**. | USD/unit product |

$$NETCOST = TACC + TOC + TADC - TREV \tag{62}$$

$$TACC = AF * \sum\nolimits_{i \in \textbf{PROCESS}} z_i * CC_i \tag{63}$$

$$AF = \frac{d * (1 + d)^n}{(1 + d)^n - 1} \tag{64}$$

$$TOC = \sum\nolimits_{i \in \textbf{PROCESS}} z_i * OC_i \tag{65}$$

$$TADC = \sum\nolimits_{i \in \textbf{DISP}} FPI_i * DC_i * DPY \tag{66}$$

$$TREV = \sum\nolimits_{i \in \textbf{REVP}} FPI_i * SP_i * DPY \tag{67}$$

$$FPI_i = \sum\nolimits_{s \,\in\, \textbf{STF}_i} \sum\nolimits_{c \,\in\, \textbf{FPCO}_i} \sum\nolimits_{k \,\in\, \textbf{P}_i} X_{k,i}^{s,\,c}, \ \ \forall\, i \in \textbf{PRODUCT} \tag{68}$$

## 3. Case Study

### 3.1. Case Study Parameters

The parameters in the mathematical model formulation discussed above were given appropriate values for the purpose of performing a case study for a sludge treatment plant. Those values were either reasonably assumed or extracted from various sources to help illustrate the applications of the developed optimization model. Table 12 lists the feed property-related parameters, while Table 13 lists the capital and operating costs for each technology. Moreover, Table 14 presents the selling prices and disposal costs of the final products, and finally, Table 15 lists the process performance-related parameters. As the capital and operating costs were gathered from different sources in the literature that vary in currency and year of study, those values were adjusted for inflation, using the Chemical Engineering Plant Cost Index (CEPCI) [53], for the year 2019, and were then converted

to US dollars (USD/$) for consistency. The economies of scale exponent $\alpha_i$ was assumed to be 0.6 for all the technologies. The case study was evaluated for a project lifetime ($n$) of 20 years at an interest/discount rate ($d$) of 7.5% [15]. Continuous chemical processing plants typically operate 8000 h per year; hence, the $DPY$ parameter was assumed to be 333 days/year.

**Table 12.** Feed Property Parameters.

| Parameter | Value | Units |
|:---:|:---:|:---:|
| $FTHS$ | 100 | tDS/day |
| $FPVS$ | 70 | % |
| $FPASH$ | 30 | % |
| $PDS_{THS}$ | 5 | % |

**Table 13.** Capital and Operating Costs of Technologies.

| Technology | $BCC_i$ (MMUSD) | $BQ_i$ (tDS/day) | $POC_i$ (USD/tDS) | Ref. |
|:---:|:---:|:---:|:---:|:---:|
| MAD | 31.86 | 100 | 52 | [54] |
| MADT | 33.26 | 100 | 62 | [54] |
| CD | 2.16 | 50 | 58 | [35] |
| CU | 2.16 | 50 | 58 | [35] |
| BPD | 6.6 | 50 | 69 | [35] |
| BPU | 6.6 | 50 | 69 | [35] |
| FPD | 8.2 | 50 | 134 | [35] |
| FPU | 8.2 | 50 | 134 | [35] |
| TD | 12.59 | 480 * | 26 ** | [55] |
| INC | 34.62 | 130 | 95 | [56,57] |
| GN | 2.09 | 5 | 154 | [58] |
| PY | 8.26 | 50 | 100 | [57] |
| SCO | 9 | 14 | 113 *** | Correspondence with SCFI [59] |
| SCG | 18.44 | 24 | 175 | [32] |

* tH$_2$O(evaporated)/day; ** USD/tH$_2$O (evaporated) ; *** USD/tVS.

**Table 14.** Final Product Disposal Costs and Selling Prices.

| Final Product | $DC_i$ (USD/ton) | $SP_i$ (USD/ton) | Ref. |
|:---:|:---:|:---:|:---:|
| DS20 | 250 | N/A | [56] |
| DS40 | 125 | N/A | [56] |
| ASH | 77 | N/A | [56] |
| E | N/A | 0.08 | [60] |
| FERT | N/A | 30 | [61] |
| BO | N/A | 285 * | [62] |
| BC | N/A | 200 | [63] |
| H2 | N/A | 2 ** | [64] |

* Assuming a price equivalent to 70% of crude oil of price $\approx$ 60 USD/bbl. ** USD/kg.

**Table 15.** Technology Performance-Related Parameters.

| Parameter | Value | Units | Ref. |
|:---:|:---:|:---:|:---:|
| $VSD_{MAD}$ | 50 | % | [54] |
| $Y^E_{MAD}$ | 2390 | kWh/tVSD | [54] |
| $VSD_{MADT}$ | 60 | % | [54] |
| $Y^E_{MADT}$ | 2390 | kWh/tVSD | [54] |
| $DR^P$ | 0.004 | ton/tDS | [34] |
| $DR^L$ | 0.1 | ton/tDS | [34] |
| $DR^{FC}$ | 0.07 | ton/tDS | [34] |
| $PDS_{CU}, PDS_{CD}$ | 10 | % | [35] |
| $PDS_{BPU}, PDS_{BPD}$ | 20 | % | [35] |
| $PDS_{FPU}, PDS_{FPD}$ | 40 | % | [35] |
| $PDS_{TD}$ | 90 | % | Typical |
| $LHV_{VS}$ | 21,000 | MJ/tVDS | [65] |
| $\lambda_W$ | 2260 | MJ/tonne | Steam Table |
| $HLF$ | 0.05 | Dimensionless | assumed |
| $\eta_R$ | 25 | % | [57] |
| $CF_{MJ2kWh}$ | 0.27778 | Dimensionless | |
| $YF^E_{GN}$ | 1368 | kWh/tVS | [58] |
| $CF^{BO}_{PY}$ | 1 | Dimensionless | |
| $CF^{BC}_{PY}$ | 1 | Dimensionless | |
| $YF^E_{SCO}$ | 825 | kWh/tVS | Correspondence with SCFI [59] |
| $YF^{H2}_{SCG}$ | 112 | kgH2/tVS | [32] |

*3.2. Sensitivity Analysis*

The model economic parameters used in this case study were subject to several sources of uncertainty rooting from: (1) the inconsistent basis for factors used in capital cost calculations, (2) the assumption that the operating costs varied linearly with the processing capacity, (3) volatility of products selling prices and the market demand, and (4) uncertainty in possible government incentives for each technology. The model technical performance-related parameters were also prone to a level of uncertainty due to: (1) the infancy of some of the new developed technologies, (2) different sludge characteristics that the original sources relied on, (3) scalability issues, etc. Consequently, the need to assess the model sensitivity to each of the relevant uncertain parameters was a necessity. The capital and operating costs of each technology were usually estimated from preliminary techno-economic studies for feasibility purposes. This type of study corresponded to a Class 4 cost estimate as defined by the Association for the Advancement of Cost Engineering (AACE) and could have an accuracy between +50% and −30% [66]. This range was used for the sensitivity analysis on capital and operating costs parameters.

In this study, selling prices were assessed for the following ranges: Electricity price from 6 to 30 cents per kWh, fertilizer price from 20 to 100 USD/ton, bio-oil from 100 to 500 USD/ton (15 to 70 USD/bbl of bio-oil), biochar from 100 to 500 USD/ton, and H$_2$ from 1 to 5 USD/kg. Disposal costs of dewatered sludge and ash varied from 25 to 75 USD/wet ton of solids and from 40 to 100 USD/ton of ash, respectively. The discount rate also varied between 5% and 10%. As far as performance-related parameters were concerned, yield parameters and sludge LHV varied by ±30%. The percentage of dry solids produced from belt press and filter press dewatering varied between 12–37% and 27–46%, respectively; these ranges covered the whole spectrum of dewatering efficiencies

for those two technologies to account for extreme cases of sludge composition variations. Finally, the feed characteristics were examined as follows: inlet flowrates from 50 tDS/day to 150 tDS/day and composition of sludge VS% from 50% to 80% with a corresponding Ash% of 50% to 20%. In this study, the results of the sensitivity analysis are reported to understand how a change in objective function value was accompanied by a change in the optimal processing pathway.

### 3.3. Results and Discussion

#### 3.3.1. Base Case Results

The proposed optimization model formulation together with the case study parameters values were entered into GAMS software. The BARON solver [67] was used to solve the MINLP model guaranteeing global optimality within a reasonable runtime. The model and solver characteristics are summarized in Table 16.

**Table 16.** Model and Solver Characteristics.

| Model Statistics | | Solver Statistics | |
|---|---|---|---|
| Single Equations | 635 | Solver | BARON |
| Single Variables | 418 | Optimality Tolerance | $10^{-6}$ |
| Non-linear matrix entries | 371 | Branch-and-reduce iterations | 41 |
| Discrete Variables | 14 | Max. no. of nodes in memory | 21 |
| Non-zero elements | 1700 | CPU Time (s) | 70.72 |

The optimal processing pathway was determined by looking at the results of both the discrete variable $z_i$ and the continuous variable $SF^s_{i,j}$, where the former states the choice of a certain technology, and the latter foresees whether a certain technology product stream is split between more than one destination. The non-zero z variables obtained in the solution were for process identifiers *FPU*, *TD*, *PY* with the final products being bio-oil and biochar as per the split factor results. Figure 4 shows a schematic for the optimal processing route with stream flowrates.

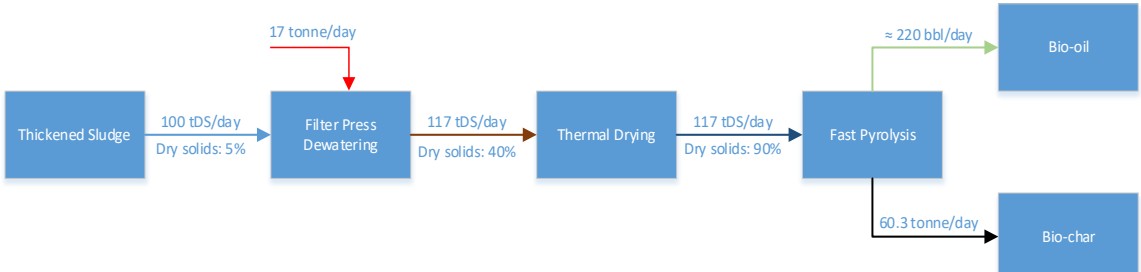

**Figure 4.** Optimal Solution Pathway with product flowrates.

The annual net cost for this pathway was approximately 6 million USD/year for a daily load of 100 tDS of sludge, while the specific cost of sludge treatment came to 180 USD/tDS. This cost of treatment per ton of dry sludge was in the same order of magnitude with reported ranges between 100 and 800 USD/tDS of various conventional sludge handling methods (i.e., landfilling, land application, and incineration) [68]. This implies that the parameters used for the case study were reliable enough for demonstrating the applicability of the proposed optimization model. The annual costs were close to 13 million USD/year where 75% of that cost was attributed to the operating costs of the different technologies and the remaining 25% was related to the annualized capital cost payments. The annual revenues were 7 million USD/year with bio-oil sales contributing to 43% of the total revenue, and a biochar share of 57%. Figure 5a shows the revenues from product sales of

the selected pathway in comparison to the total costs. Figure 5b illustrates the breakdown of costs between the different technologies in the selected pathway. It is worth mentioning that the operating cost of the filter press dewatering process accounted for the highest portion of the total costs with 34%, followed by pyrolysis operating costs with 30%. The capital cost of the dewatering step was also comparable to that of the pyrolysis. This shows how significant and important the dewatering step is in the whole processing route.

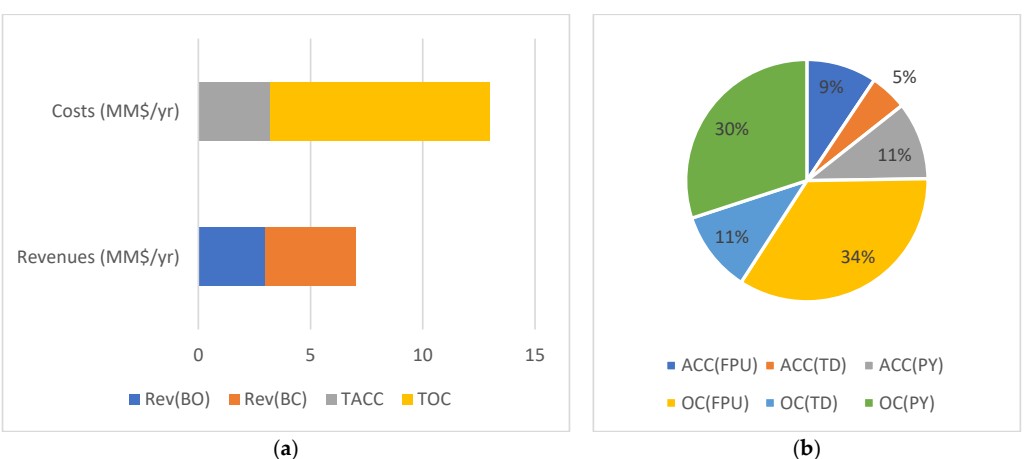

| (**a**) | (**b**) |

**Figure 5.** (**a**) Optimal Pathway Costs and Revenues (**b**) Optimal Pathway Costs Breakdown.

3.3.2. Sensitivity Analysis Results

Feed Characteristics

- Feed Flowrate

The changes of feed flowrate in the studied ranges, from 50 to 150 tDS/day, did not have any impact on the optimal processing route which was selected in the base case scenario. Nevertheless, there were obvious changes in the value of the objective function or the net cost value as presented in Figure 6. To see the effect of economies of scale, the percentage of net cost increase per a 10 tDS/day rise in feed flowrate was added in the same plot. As expected, the percentage of additional net annual costs decreased with increasing the capacity from 15% per each extra 10 tDS/day at 60 tDS/day capacity, to 5.7% at 150 tDS/day.

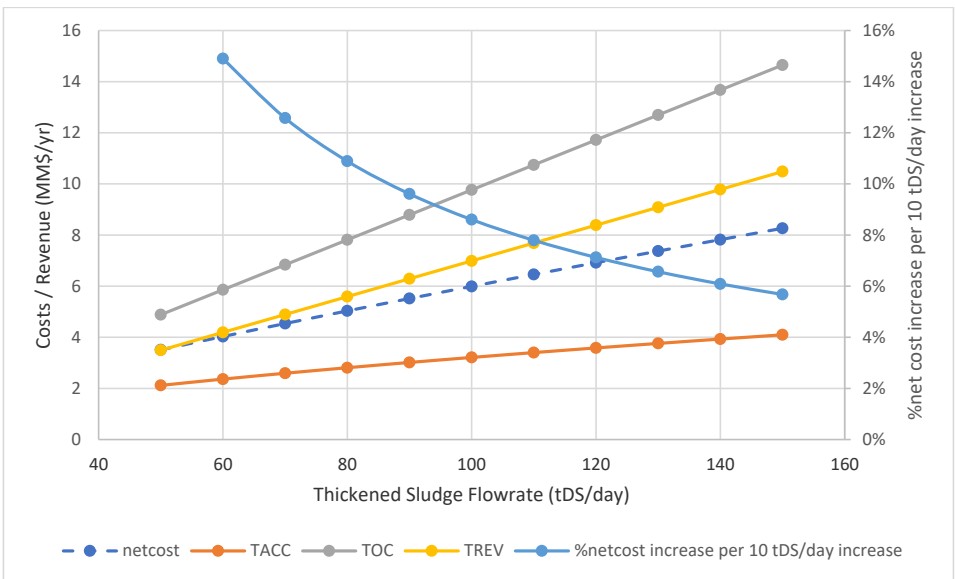

**Figure 6.** Sensitivity Analysis of Economic Variables (MMUSD/yr) with Feed Flowrate.

However, it is not yet clear from the above results which of the net cost components (i.e., **TACC**, **TOC**, and **TREV**) had the most underlying impact on the net cost reduction. Therefore, another method to examine the capacity effects is presented, which aimed to conduct the comparison against the various economic variables, but per unit ton of dry sludge treated. The specific costs/revenue variables are suffixed by the asterisk symbol (*) and were plotted against different feed flowrate values as shown in Figure 7. It was observed that the specific operating costs and revenues in USD/tDS were constant across the whole spectrum of capacities studied at values of 293 USD/tDS and 210 USD/tDS, respectively. Thus, these two components did not play a significant role in the economies of scale. On the other hand, the annualized capital costs per unit of sludge exponentially reduced with increasing capacities. Hence, they were the sole drivers behind the changes in the specific net cost results. The rate of change in specific annual capital costs, and accordingly, that of specific net costs, decelerated with increasing capacity from a 4% reduction per each extra 10 tDS/day of feed flowrate at an initial capacity of 60 tDS/day to 1.5% at 150 tDS/day. This indicates that the effects of scaling economies are minimal at capacities higher than 150–200 tDS/day and that the specific net costs will asymptote at values close to 155–160 USD/tDS.

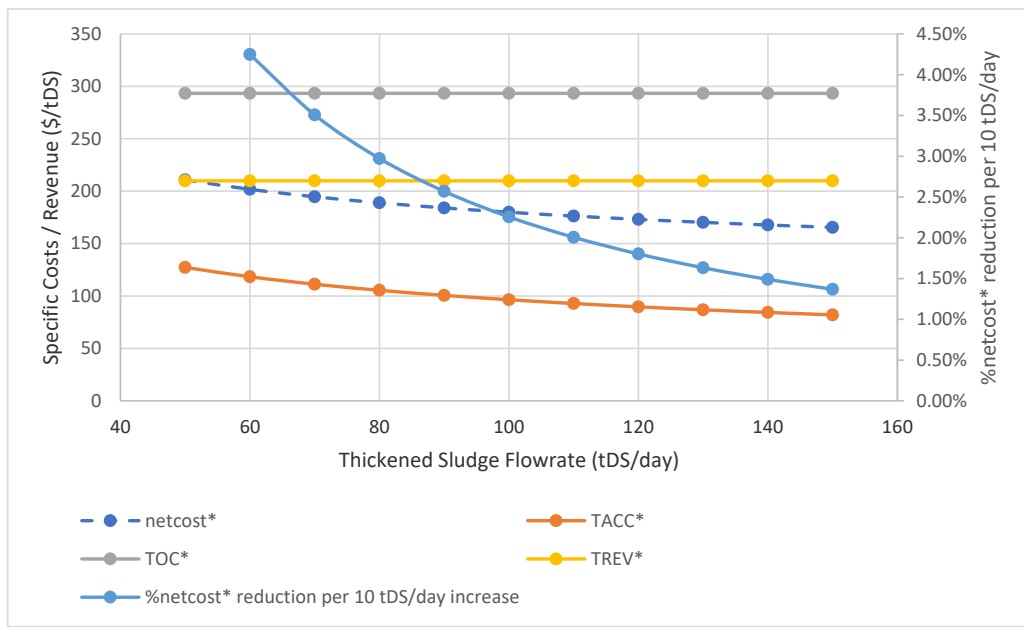

**Figure 7.** Sensitivity Analysis of Specific Economic Variables (USD/tDS) with Feed Flowrate.

-       Feed Composition

The impact of changing the composition of the sewage sludge on the objective function was seen to be minimal. As shown in Figure 8, a 5% increase in the sludge volatile solids percentage only led to a reduction of approximately 0.65% in the net cost primarily due to the revenue increases from higher bio-oil yields at the expense of biochar. Thus, the maximum variation expected in the net cost for a composition change of 30% (i.e., from 50% to 80%) is close to 3.8% which is not trivial if compounded annually; however, it does not undermine the feasibility of the processing route if a certain wastewater treatment facility is generating sludge with lower organic contents. The optimal processing pathway did not change with varying the composition and this is visible also in the figure from the constant capital and operating costs that were dependent mainly on the total amount of dry solids, regardless of their components analysis for the chosen processing route.

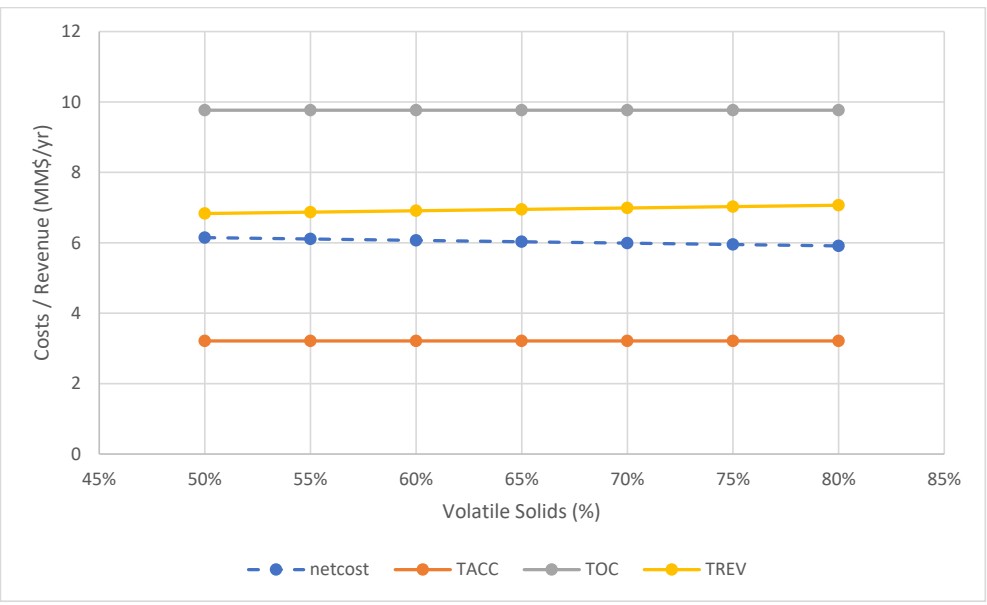

**Figure 8.** Sensitivity Analysis of Economic Variables (MMUSD/yr) with Feed Composition.

Economic Parameters

- Capital and Operating Costs

The objective function value is sensitive to any capital and/or operating cost variations of all the technologies present in the optimal pathway of the base case, namely: *FPU*, *TD*, and *PY*. The percentage of change in capital cost and operating cost parameters of those technologies were plotted against the % change in the objective function (compared to base case results) and demonstrated in Figures 9 and 10, respectively. The objective function value plateaued after an increase greater than 20% for the pyrolysis operating cost; this was because at such value the optimization model decides to discard pyrolysis technology from the optimal pathway and the thermally dried biosolids are chosen to be sold as fertilizers instead of being further processed. Similarly, with an increase in *FPU* operating costs higher than 10%, the net cost stagnates and the optimal pathway changes to *BPU*, *TD*, *PY*. Reductions in *FPU* operating costs had the most significant impacts on net cost where at −30%, the corresponding decrease in objective function value was 22%.

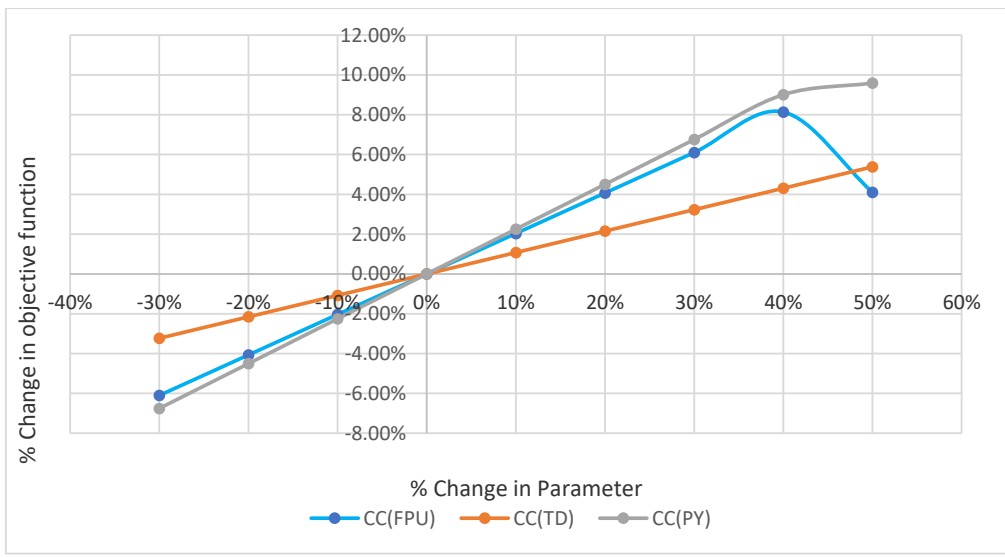

**Figure 9.** Sensitivity Analysis of Optimal Pathway Technologies' Capital Cost Parameters.

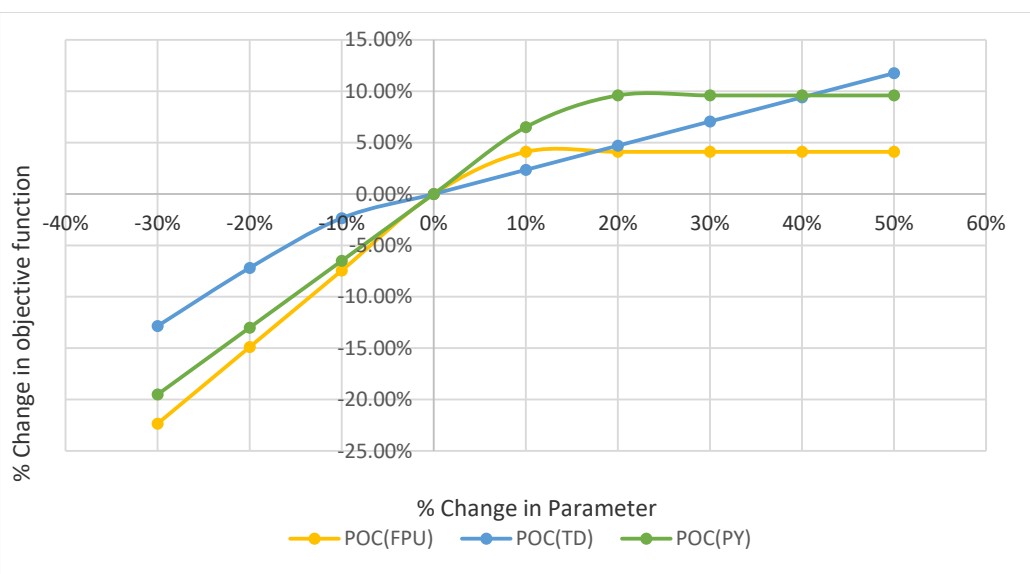

**Figure 10.** Sensitivity Analysis of Optimal Pathway Technologies' Operating Cost Parameters.

*MADT* was the only technology outside the base case optimal pathway that the capital cost parameter of which had an impact on the net cost (objective function). This impact appeared only at a 30% reduction in *MADT* capital cost, which led to a 4% decrease in the net cost, and a different optimal pathway, as shown in Figure 11 where the thickened sludge underwent thermal hydrolysis pre-treatment and anaerobic digestion before eventually being sent to the pyrolysis pathway of the base case. This adds an additional energy product in the form of electricity recovered from MAD biogas.

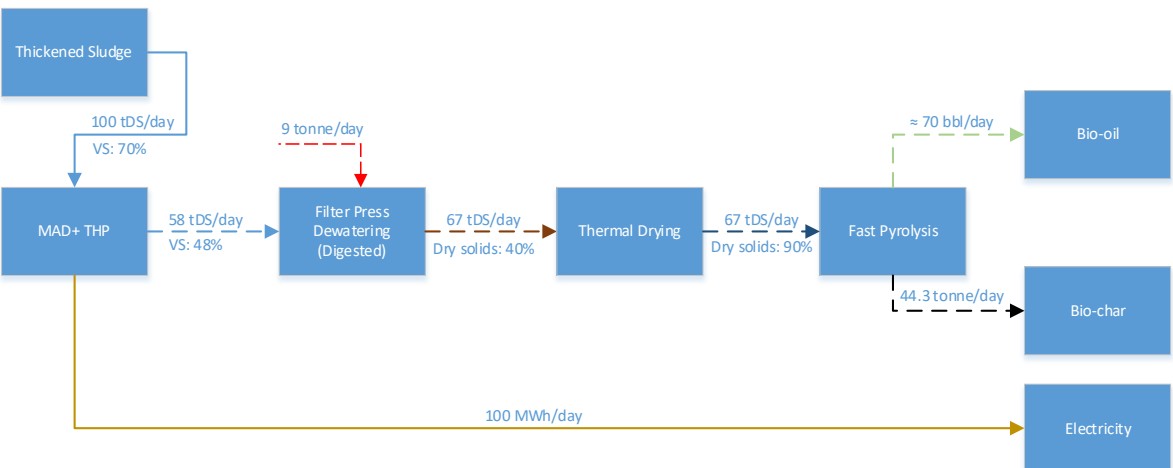

**Figure 11.** Optimal Pathway at 30% Reduction in MADT Capital Cost.

As far as the remaining technologies' operating cost parameters are concerned, at −30% for *FPD*, a slight reduction in the net cost of approximately 1% can be attributed to a corresponding optimal processing pathway similar to that presented in Figure 11. Furthermore, reductions in operating costs of *BPU* by 20% and 30% corresponded to the objective function value dropping by 7% and 3.5%, respectively. The resulting optimal pathway was similar to that of the base case, except that *FPU* was replaced by *BPU*, which produced a sludge of 20% solids in comparison to the 40% solids produced by *FPU*. The remaining operating cost parameters for the other technologies were found to be insensitive to both the objective function as well as the optimal pathway choice.

- Product Selling and Disposal Prices

All the final products' selling prices had an impact on the objective function value when compared to the base case. Electricity prices appeared to be the most sensitive parameter compared with the remaining final products. The highest price studied for electricity (30 cents/kWh) brought the net cost down by −130% compared to the base case results, which actually led to having a net profit at such a rate (Figure 12). The optimal pathway chosen at this electricity price was *MADT*, *BPD*, *GN*. Such a price was much higher than the average prices for industrial use, thus accounting for more optimistic scenarios of government incentives to electricity from waste such as feed-in-tariff (FIT) and/or tax credits policies. Followed by electricity, the selling prices of biochar and hydrogen were the second most sensitive to changes. At their higher limits (500 USD/ton of biochar and 5 USD/kg $H_2$), they caused a reduction close to 100% of the objective function value compared to the base case scenario. The optimal pathway for biochar's highest price stayed the same as the base case, while for $H_2$ it changed to *CU + SCG* even at prices as low as 3 USD/kgH$_2$. This shows the promising potential of that technology, especially with the future higher demands of a sustainable hydrogen economy. The objective function value was least sensitive to the prices of bio-oil and Class A biosolids fertilizer relative to the remaining products. However, significant reductions of approximately 40% to the net costs were achievable at the higher end of the studied price range.

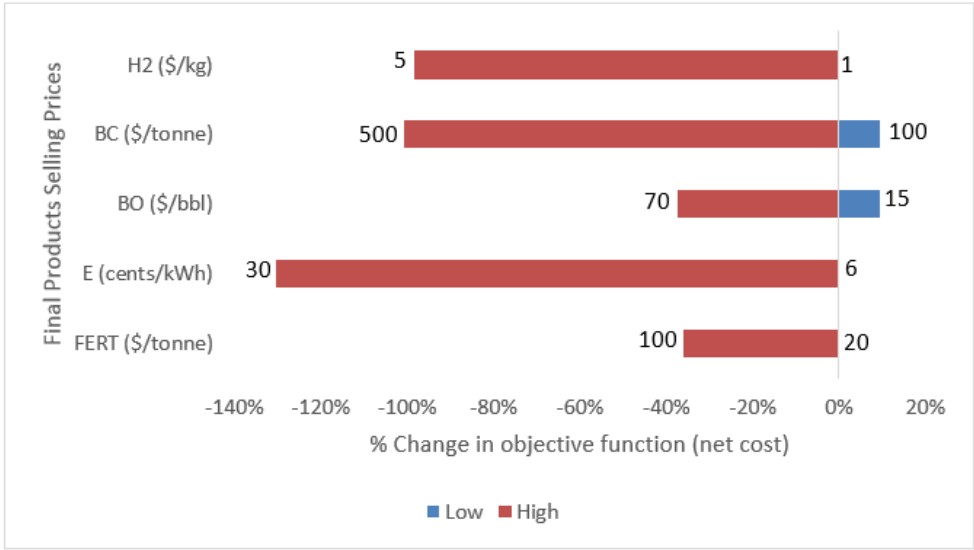

**Figure 12.** Sensitivity Analysis of Final Product Selling Prices.

On the other hand, changes in the disposal costs of the by-products (i.e., dewatered sludge, ash) did not impact on either the optimal pathway selected by the model nor the objective function value.

- Discount Rate

The objective function value was sensitive to variations of the discount rate *d*; however, the optimal pathway stayed the same. As shown in Figure 13, an incremental change of ±0.5% in *d*, led to a ±2% change in the objective function, which is a significant change. This suggests that at higher rates of inflation trends, the investment in such projects can be less attractive without further incentives.

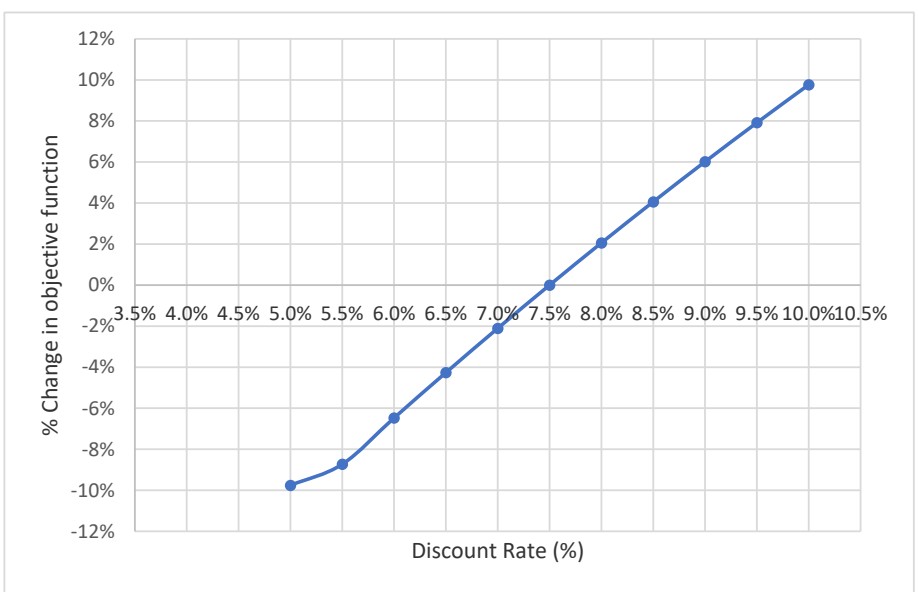

**Figure 13.** Sensitivity Analysis of the Discount Rate.

Performance-Related Parameters

Changing the yield parameters of products of ***MAD***, ***MADT***, ***GN***, ***INC***, ***SCO***, and ***SCG*** by ±30% had no effect on both the objective function and optimal processing pathway. This indicates the robustness of the pyrolysis pathway against a wide range of process efficiencies of the competing technologies. As far as pyrolysis product yields are concerned, an inverse proportion relationship existed between them and the objective function value. As shown in Figure 14, a 10% increase in bio-oil yield caused a 5% decrease in net cost and vice versa. However, at bio-oil yield reductions below 20%, the optimal pathway changed to only filter press dewatering followed by thermal drying, where the dried sludge could be sold as fertilizer causing no more additional reduction in net costs. A similar relationship existed between biochar yield and the objective function value. However, a 10% increase in the yield caused a 6.7% reduction in the net cost. Biochar yield reduction increased the net cost also by 6.7% until the objective function reduction value stagnated at any yield reduction below 15% as shown in Figure 15. This indicates that the objective function value was more sensitive to changes in the biochar yield than bio-oil which was expected based on the sensitivity analysis results of products' selling prices as it was discussed earlier.

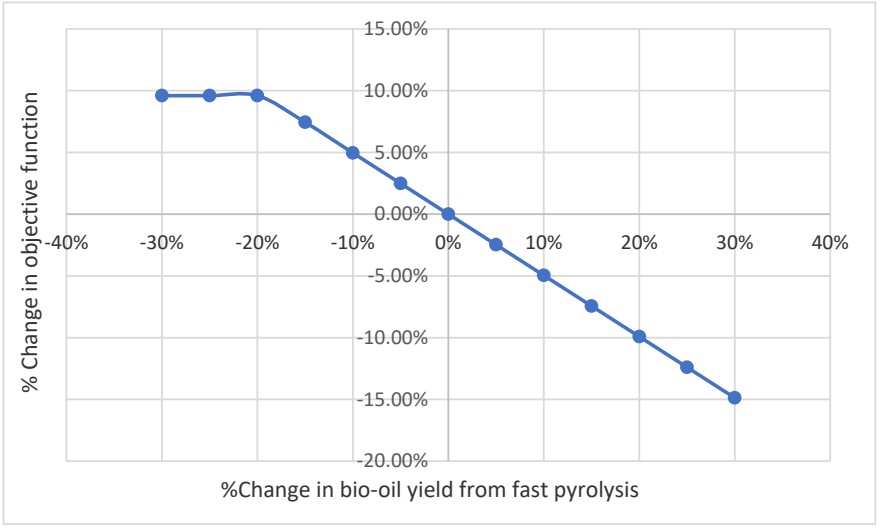

**Figure 14.** Sensitivity Analysis of the Bio-oil Yield Parameter.

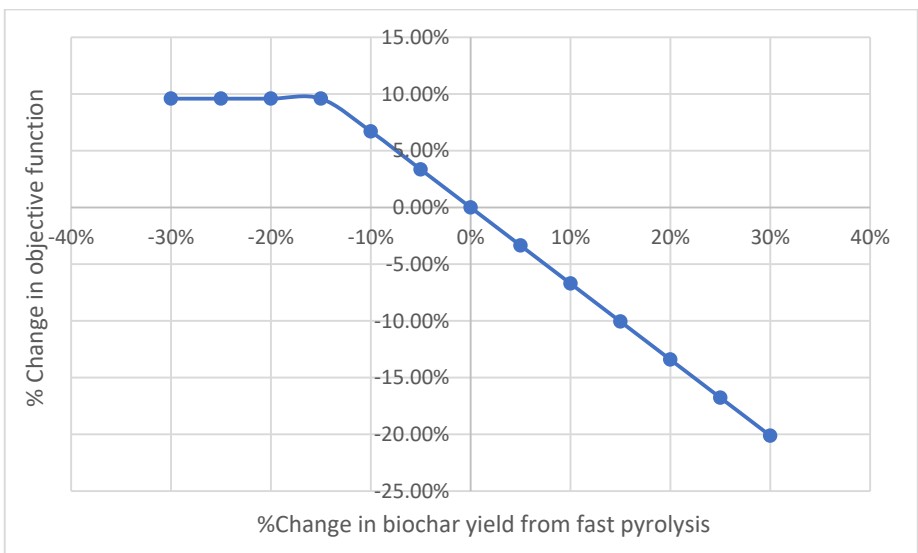

**Figure 15.** Sensitivity Analysis of the Biochar Yield Parameter.

The efficiency of the dewatering processes was found to have an impact on the results. For belt dewatering, the assessed values were between 12% and 37% with a 5% increment. There was no change in results for dry solids% up to 22%. Starting from 27%, the optimal pathway favored a pathway of *BPU* followed by *TD* with the final product being Class A biosolids sold as fertilizers. Results for the latter three scenarios are listed in Table 17.

**Table 17.** Sensitivity analysis results for belt press dewatering efficiency runs.

| $PDS_{BP}$ **(Cases)** | **22% (and Lower)** | **27%** | **32%** | **37%** |
|---|---|---|---|---|
| **Optimal Pathway** | *FPU + TD + PY* | *BPU + TD* | *BPU + TD* | *BPU + TD* |
| *TACC* **(MMUSD/yr)** | 3.21 | 1.84 | 1.72 | 1.62 |
| *TOC* **(MMUSD/yr)** | 9.77 | 4.54 | 4.03 | 3.67 |
| *TREV* **(MMUSD/yr)** | 6.99 | 1.00 | 1.00 | 1.00 |
| *NETCOST* **(MMUSD/yr)** | 5.99 | 5.37 | 4.75 | 4.28 |

The percentage of dry solids produced from filter press dewatering varied between 27% and 48% with increments of 4%. The objective function value changed for all the assessed values. Optimal processing pathway swapped *FPU* in the base case results with *BPU* at outlet dry solids values between 27% and 35%. From 39% to 48%, the same pathway as that of the base case remained unchanged. Table 18 shows the summary of economic parameter values at those different values.

**Table 18.** Sensitivity analysis results for filter press dewatering efficiency runs.

| $PDS_{FP}$ **(Cases)** | **35% (and Lower)** | **39%** | **43%** | **48%** |
|---|---|---|---|---|
| **Optimal Pathway** | BPU + TD + PY | FPU + TD + PY | FPU + TD + PY | FPU + TD + PY |
| *TACC* **(MMUSD/yr)** | 3.30 | 3.23 | 3.16 | 3.10 |
| *TOC* **(MMUSD/yr)** | 9.01 | 9.83 | 9.59 | 9.39 |
| *TREV* **(MMUSD/yr)** | 6.08 | 6.99 | 6.99 | 6.99 |
| *NETCOST* **(MMUSD/yr)** | 6.24 | 6.07 | 5.76 | 5.50 |

It can be concluded from the results of both dewatering processes that more efficient belt press dewatering can yield higher cost savings compared to filter press dewatering assuming the same capital and operating costs with the higher efficiency.

The last studied performance-related parameter was the LHV value of the sludge, which could potentially have an impact on favoring the incineration technology at higher values. Nevertheless, even at an increase of 30% of LHV value, neither the objective function value nor the chosen optimal pathway changed.

### 3.3.3. Complementary Analysis

It can be observed from the above results that the following technologies were not selected in any of the studied scenarios under the sensitivity analysis: *MAD*, *INC*, and *SCO*. Hence, additional runs were performed on GAMS software for further investigation. In each run, the binary variable $z_i$ of one of these technologies was forced to equal 1, while all other conversion technologies $z_i$ values were set to 0. Table 19 lists the processing pathway resulting from each run as well as the change in the different components of the objective function compared to the base case scenario results. It is clearly evident from the results that the main driver for the lower objective function value of the base case scenario in comparison to the other three pathways was the higher annual revenues from selling biochar and bio-oil compared to electricity. In addition, there were no additional disposal costs required for pyrolysis products compared to the dewatered sludge from *MAD* and ash from *INC* and *SCO* that required transportation and disposal expenses. The operating and maintenance cost of all three pathways were significantly lower when compared to the base case; however, this offset the remaining objective function components.

**Table 19.** Additional Runs Results Summary.

| Processing Pathway | Δ TACC | Δ TOC | Δ TADC | Δ TREV | Δ NETCOST |
|---|---|---|---|---|---|
| *MAD* + *FPD* | +0.80 MMUSD/yr [+25%] | −5.13 MMUSD/yr [−52%] | +3.17 MMUSD/yr [N/A] | −4.76 MMUSD/yr [−68%] | +3.65 MMUSD/yr [+61%] |

### 4. Conclusions

The main objective of this study was to propose a decision-making support tool for choosing the most economic pathway of sludge-to-energy technologies via superstructure optimization techniques. A literature review was conducted to assess the state of research in the addressed problem and the gaps were identified in utilizing mixed-integer optimization for sludge-to-energy decision-making frameworks. A mathematical model customized for the problem was formulated and its applicability was tested via a case study for a hypothetical treatment facility with a capacity of 100 tDS/day.

One of the main outcomes from the case study was that although the proposed model is a MINLP formulation, which is usually difficult to solve, global optimal solutions were found efficiently within a reasonable computing time. The base case results showed that a combination of filter press dewatering followed by thermal drying and fast pyrolysis was deemed to be the most economical pathway among the available alternatives. The products of such a pathway were bio-oil, which can be used as an alternative fuel upon refining, and biochar, which has a variety of useful applications such as an adsorption material or in agriculture. The estimated net specific cost for processing a ton of dry sludge using this pathway was USD 180, which is in the same order of magnitude of current ranges of conventional sludge handling methods, but with added environmental and social benefits.

The parameters used in the case study were extracted from various sources in the literature and vendor documents. So, the model parameters defined in the case study were subjected to a high degree of uncertainty related to the reliability and availability of high-quality data in the literature, as well as uncertainties related to the market volatility of the final product prices and government subsidies or incentives that can be provided to such products. Therefore, a sensitivity analysis for wide expected ranges of each of those uncertain parameters was conducted to determine the impact of each individual parameter variation on the objective function value represented by the annual net costs

and/or changes in the optimal pathway that is selected by the model. In this regard, the following results were achieved:

- The technology selection route was sensitive to the capital cost parameter of *MADT*, and operating costs of *FPD* and *BPU*. Changes in the remainder of the technologies' capital and operating cost parameters did not impact the model outputs;
- Variations in the final products' prices also had a significant effect on the selected optimal pathway and the net costs of the selected plant. Electricity price was the most sensitive parameter followed by hydrogen and biochar prices, while bio-oil and Class A biosolids (fertilizer) prices were found to have the least relative effect on the objective function values;
- The objective function values were also sensitive to the value of discount rates; however, the technology selection did not change with reasonable interest variations;
- Changing the yield parameters of technologies other than fast pyrolysis had no influence on the solution. This indicates the robustness of the pyrolysis pathway against a wide range of process efficiencies of the competing technologies;
- The objective function was highly sensitive to all the parameters related to the technologies in the base case optimal pathway, which proved the applicability of the proposed model and provided sensible results;
- The feed characteristics affected the optimal cost value, which was explained by the economies of scale. The inverse relationship between net cost and process capacity effects started to diminish at capacities above 200 tDS/day. There were slight impacts from changing the composition of the sewage sludge where cost reductions were observed at a higher %VS due to an increased yield of energy products correlated with an increase in the organic contents of the sludge.

Finally, it should be noted that although the optimization framework proposed in this study can be used as an early screening tool for decision-makers to assess different sludge-to-energy pathways, it can be further extended to account for different feedstocks and also to consider the effects of environmental constraints ($CO_2$ emissions).

**Author Contributions:** Methodology, O.M.; Software, O.M.; Supervision, Q.Z., A.A. and A.E.; Writing—original draft, O.M. and F.H.; Writing—review and editing, F.H., Q.Z. and A.E. All authors have read and agreed to the published version of the manuscript.

**Funding:** This research received no external funding.

**Data Availability Statement:** The data presented in this study are available on request from the corresponding author.

**Conflicts of Interest:** The authors declare no conflict of interest.

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
