# Peer review of "A Superstructure Mixed-Integer Nonlinear Programming Optimization for the Optimal Processing Pathway Selection of Sludge-to-Energy Technologies"

_sustainability, doi:10.3390/su15054023_

Round 1
Reviewer 1 Report
This paper presents an important research on optimization of processing pathways of sludge to energy technologies. The research is valuable to achieve sustainable waste water treatment. A thorough research was carried out to explore the technologies from economic perspective. The results from this work can provide useful guidelines for plant development. The pape is generally well written. There are some minor errors that should be corrected, e.g.,
Page 2, line 1, the sentence “… which are practices gaining less social and legislative support…” should be revised.
Page 2, line 3, the sentence “ … is increasingly important than ever” should be revised
Page 7, at 2nd line to the bottom of the page, there is error that needs to be corrected.
Page 8, in section 2.2, there is an error of paragraph break in the first paragraph.
In addition, equations are not lined up properly and should be revised. In Fig. 1 and Fig. 2, the color of shades and fonts color can be revised to improve clarity.
Author Response
Point 1: This paper presents an important research on optimization of processing pathways of sludge to energy technologies. The research is valuable to achieve sustainable waste water treatment. A thorough research was carried out to explore the technologies from economic perspective. The results from this work can provide useful guidelines for plant development. The paper is generally well written. There are some minor errors that should be corrected, e.g.:
Page 2, line 1, the sentence “… which are practices gaining less social and legislative support…” should be revised.
Page 2, line 3, the sentence “ … is increasingly important than ever” should be revised
Page 7, at 2nd line to the bottom of the page, there is error that needs to be corrected.
Page 8, in section 2.2, there is an error of paragraph break in the first paragraph.
In addition, equations are not lined up properly and should be revised. In Fig. 1 and Fig. 2, the color of shades and fonts color can be revised to improve clarity.
Answer to Reviewer #1:
Done. Thorough typos and grammar checks have been done over the manuscript.
All the mentioned errors have been corrected.
All equations have been properly lined up.
Some figures have been modified to improve their quality.

Reviewer 2 Report
In this paper the authors provide a brief overview of a set of the most promising sludge-to-energy conversion technologies. Also, a mathematical model is developed, using a superstructure optimization-based approach as a decision-making support tool. This paper does not appear to be a research article. It is too informative and abstractly written. It is most like a review paper. Experimental results are lacking for scientific support. Intensive editing is required before publication.
Synchronize references based on instructions for authors.
Pay attention throughout the entire text to "Error! Reference source not found."
Author Response
Point 1: In this paper the authors provide a brief overview of a set of the most promising sludge-to-energy conversion technologies. Also, a mathematical model is developed, using a superstructure optimization-based approach as a decision-making support tool. This paper does not appear to be a research article. It is too informative and abstractly written. It is most like a review paper. Experimental results are lacking for scientific support. Intensive editing is required before publication.
Synchronize references based on instructions for authors.
Pay attention throughout the entire text to "Error! Reference source not found."
Answer to Reviewer #2:
All “Error! Reference source not found” statements have been corrected.
All references have been synchronized based on instructions for authors.
Moreover, in this paper, a decision-making support tool is proposed to help in choosing the optimal pathway for the sludge-to-energy conversion from a techno-economic perspective. The conversion technologies under study are: 1) anaerobic digestion, 2) pyrolysis, 3) gasification, 4) incineration, 4) supercritical water oxidation, 5) supercritical water gasification, as well as the corresponding dewatering and drying methods for each technology. So, it is necessary that we briefly talk about those technologies in the “Introduction”, while referring the interested audiences to the relevant references. In Section 2 we have thoroughly discussed the introduced methodology and in Section 3 we have presented our case study and the obtained results, applying a very reliable solver (Baron Solver which is available in GAMS software) which is generally used to evaluate the optimization techniques in the design stage of industrial applications, prior to any experimental test. Meanwhile in that section, different synergies between the available technologies are compared by the formulation of a superstructure optimization problem expressed in a mixed-integer non-linear program (MINLP) framework. Furthermore, since the model parameters are subject to uncertainty, a comprehensive sensitivity analysis section has been included. As a result, the authors still believe that the given explanations in every section of this paper are helpful for the readers to better understand the proposed methodology and the followed pathway. However, if the reviewer still persists that some parts/paragraphs are not required, we can omit them.

Round 2
Reviewer 2 Report
Authors should pay attention to the warning "Error! Source references
not found." and to correct references! Page 7, Page 16... (check all manuscript)
Author Response
Point #1: Authors should pay attention to the warning "Error! Source references
not found." and to correct references! Page 7, Page 16... (check all manuscript).
- The mentioned warnings on "Error! Source references not found" at pages 7, and 16 have been corrected. Moreover, the manuscript has been fully checked for other kinds of errors.